# Learning Representations for Hierarchies with Minimal Support

**Benjamin Rozonoyer**[1]   **Michael Boratko**[2]   **Dhruvesh Patel**[1]   **Wenlong Zhao**[1]
**Shib Dasgupta**[1]   **Hung Le**[1]   **Andrew McCallum**[1]
[1]University of Massachusetts Amherst
[2]Google DeepMind
{brozonoyer,dhruveshpate,wenlongzhao,ssdasgupta,hungle,mccallum}@cs.umass.edu
mboratko@google.com

## Abstract

When training node embedding models to represent large directed graphs (digraphs), it is impossible to observe all entries of the adjacency matrix during training. As a consequence most methods employ sampling. For very large digraphs, however, this means many (most) entries may be unobserved during training. In general, observing every entry would be necessary to uniquely identify a graph, however if we know the graph has a certain property some entries can be omitted - for example, only half the entries would be required for a symmetric graph.

In this work, we develop a novel framework to identify a subset of entries required to uniquely distinguish a graph among all transitively-closed DAGs. We give an explicit algorithm to compute the provably minimal set of entries, and demonstrate empirically that one can train node embedding models with greater efficiency and performance, provided the energy function has an appropriate inductive bias. We achieve robust performance on synthetic hierarchies and a larger real-world taxonomy, observing improved convergence rates in a resource-constrained setting while reducing the set of training examples by as much as 99%.

## 1   Introduction

Consider the directed graph and its associated adjacency matrix in Figure 1. In situations where this adjacency matrix is sparse, we can store it more efficiently by keeping a list of only the positive entries, effectively assuming any pairs not in our list are zero. But what about situations where we cannot assume that we have observed the full graph? For example, when obtaining annotations for edges of an unknown graph, the full adjacency matrix is unknown to us and we obtain the value of any particular entry by requesting an annotation. A less obvious scenario is training a model to represent a given graph. From the model's perspective, every entry of this adjacency matrix is unknown, and it is only observed as a consequence of training. Therefore, it is of interest to determine: **what is the smallest set of entries necessary to uniquely determine the graph?**

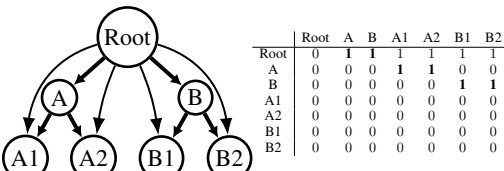

|      | Root | A | B | A1 | A2 | B1 | B2 |
|------|------|---|---|----|----|----|----|
| Root | 0    | **1** | **1** | 1  | 1  | 1  | 1  |
| A    | 0    | 0 | 0 | **1**  | **1**  | 0  | 0  |
| B    | 0    | 0 | 0 | 0  | 0  | **1**  | **1**  |
| A1   | 0    | 0 | 0 | 0  | 0  | 0  | 0  |
| A2   | 0    | 0 | 0 | 0  | 0  | 0  | 0  |
| B1   | 0    | 0 | 0 | 0  | 0  | 0  | 0  |
| B2   | 0    | 0 | 0 | 0  | 0  | 0  | 0  |

Figure 1: A transitively closed directed tree with branching factor 2 and depth 2, with the associated adjacency matrix.

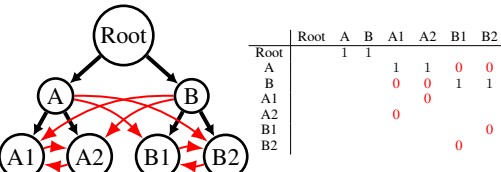

|      | Root | A | B | A1 | A2 | B1 | B2 |
|------|------|---|---|----|----|----|----|
| Root |      | 1 | 1 |    |    |    |    |
| A    |      |   |   | 1  | 1  | 0  | 0  |
| B    |      |   |   | 0  | 0  | 1  | 1  |
| A1   |      |   |   |    | 0  |    |    |
| A2   |      |   |   | 0  |    |    |    |
| B1   |      |   |   |    |    |    | 0  |
| B2   |      |   |   |    |    | 0  |    |

Figure 2: A sufficient set of entries in the adjacency matrix to uniquely distinguish the graph in Figure 1 among all transitively-closed DAGs.

38th Conference on Neural Information Processing Systems (NeurIPS 2024).

In general, the answer is "all of them", but if we assume some structural prior on the graph itself, it may be possible to reduce the number of edges necessary for consideration. For example, if we knew it was a simple graph (no self-loops) we could omit the diagonal. Similarly, for a symmetric graph, we could omit half the entries. For the class of acyclic graphs, we can omit certain entries which, were they to be 1, would form a cycle, and therefore must be 0; however, the explicit characterization of these entries is not as straightforward as the preceding cases. Focusing on the case when the graph is a transitively-closed directed acyclic graph (DAG), as in Figure 1, it is easy to see that of those entries which are 1, we can omit all but the transitive reduction, i.e. the bold edges, which corresponds to omitting the non-bold "1"s in the adjacency matrix. But what about pruning the zeros? For the graph in Figure 1, we can prove only 14 of the 49 entries in the adjacency matrix are needed to uniquely distinguish this graph among all transitively-closed DAGs. (See Figure 2.)

In this work, we first develop a general-purpose framework to identify a subset of entries required to uniquely distinguish a graph among others with some arbitrary graph property (Section 3.1). We then use this framework to construct a set of entries in the adjacency matrix sufficient to uniquely distinguish transitively-closed digraphs (Section 3.2), and prove that this reduced set is also minimal for transitively-closed DAGs (Section 3.3). We show how this can be leveraged to more efficiently train node embedding models for graph representation by defining the notion of "transitivity bias" (Section 4.1), and proving that box embeddings, a common graph representation model, have a transitivity bias (Section 4.3). We then combine these facts into a formal negative sampling procedure (Section 5) and demonstrate that box embeddings do benefit from training on this reduced set of entries (Section 6).[1] For related work in graph theory and representation learning, we refer the reader to Appendix A.

## 2   Background

We use the shorthand $[\![n]\!] \coloneqq \{1, \ldots, n\}$. A semicolon separates the main arguments of a function from parameters which are typically held constant (*e.g.*, $f(x; \mu, \sigma) = \frac{1}{\sigma\sqrt{2\pi}} e^{-\frac{1}{2}(\frac{x-\mu}{\sigma})^2}$). We may omit these secondary parameters when their values are clear from context. We represent a graph $G = (V, E)$ by its adjacency matrix $A_G \in \{0, 1\}^{N \times N}$, where $(A_G)_{u,v} = 1$ if and only if $(u{\to}v) \in E$. When it is clear from context, we omit the subscript and simply write $A$. We use arrows to represent edges: black $(a{\to}b)$ denotes a positive edge in $E$, and red $(a{\to}b)$ a negative edge in the edge complement $\overline{E}$ (see below).

### 2.1   Directed Graphs (Digraphs)

All graphs $G = (V, E)$ in this work will be finite simple[2] directed graphs (digraphs), where the edges are a subset of the complement of the diagonal, *i.e.* with $\mathrm{diag}(V) \coloneqq \{(v{\to}v) \mid v \in V\}$, we have $E \subseteq V^2 \setminus \mathrm{diag}(V) =: \mathrm{offdiag}(V)$. Given a simple digraph $G = (V, E)$, the *complement of $G$* is $\overline{G} = (V, \overline{E})$ where $\overline{E} \coloneqq \mathrm{offdiag}(V) \setminus E$. The *transitive closure of $G$* is $G^{\mathrm{tc}} = (V, E^{\mathrm{tc}})$ where $(u, v) \in E^{\mathrm{tc}}$ if and only if there exists a directed path from $u$ to $v$ in $G$. A *transitive reduction* of a digraph $G$ is a digraph $G'$ on $V$ with the fewest number of edges such that $(G')^{\mathrm{tc}} = G^{\mathrm{tc}}$. Note that a transitive reduction need not be a subgraph of $G$, and in general is not unique. If $G$ is acyclic, however, there is a unique transitive reduction, and it is also a subgraph of $G$ [Aho et al., 1972]. In this case we denote the transitive reduction $G^{\mathrm{tr}} = (V, E^{\mathrm{tr}})$.

### 2.2   Node Embeddings for Capturing Graph Structure

Given the task of modeling entities $V$ that have a known graph-theoretic structure $G = (V, E)$, a common approach is to learn a node embedding $\theta : V \to Z$ which maps a node $v \mapsto \theta(v)$ in *embedding space $Z$*. The graph structure is extracted from these geometric representations via an energy function $\mathrm{E} : V \times V \to \mathbb{R}_{\geq 0}$ which factors through $\theta$, *i.e.* there exists a dissimilarity function $h : Z \times Z \to \mathbb{R}_{\geq 0}$ such that $\mathrm{E}_\theta(u, v) \coloneqq \mathrm{E}(u, v; \theta) = h(\theta(u), \theta(v))$. For example, for undirected graphs it is common to use $Z = \mathbb{R}^D$ and $\mathrm{E}_\theta(u, v) = \|\theta(u) - \theta(v)\|$. This energy is interpreted as

---

[1]Our code and data are available at `https://github.com/iesl/geometric-graph-embedding`.

[2]A *simple* digraph is one without multiple edges or self-loops, *i.e.* the adjacency matrix contains only 0s and 1s, with 0s on the diagonal. We ignore self-loops in the graph modeled by the learned representations.

the unnormalized negative log-probability of edge existence. We seek to minimize the energy for positive edges $(u \rightarrow v)$ by learning representations for which there exists some (global) threshold $T$ such that $A_{uv} = 1$ if and only if $\mathrm{E}_\theta(u, v) \leq T$.

One reason to embed nodes is to learn representations end-to-end in conjunction with other objectives via gradient descent. In such a setting, a typical loss function for the graph structure takes the form

$$\mathcal{L}_{\text{full}}(\theta; G) \coloneqq \sum_{(u \rightarrow v) \in E} \ell^+\left(\mathrm{E}_\theta(u, v)\right) + \sum_{(u \rightarrow v) \in \overline{E}} \ell^-\left(\mathrm{E}_\theta(u, v)\right)$$

where $\ell^+$ and $\ell^-$ are referred to as the *positive* and *negative* loss functions, respectively, and the pairs in $E$ and $\overline{E}$ are referred to as *positive* and *negative* edges accordingly.

The loss function $\mathcal{L}_{\text{full}}$ has $|V|(|V| - 1)$ terms, and thus is often too computationally demanding to use for training. For digraphs which are sparse, a common workaround is to define a noise probability density function (pdf) $p_{\text{n}}$ and design a loss function $\mathcal{L}_{\text{noise}}$ which replaces the sum over negative edges with an expectation with respect to $p_{\text{n}}$. In practice, Monte Carlo sampling is used to calculate a loss $\mathcal{L}_{\text{sampled}}$ which approximates $\mathcal{L}_{\text{noise}}$.

## 3 Distinguishing Digraphs via Sidigraphs

We aim to define a new loss function with only a subset of the terms in $\mathcal{L}_{\text{full}}$ (which currently include all entries of the adjacency matrix $A_G$), while having the same minimizer. Thus, we first attempt to determine the minimal number of entries in $A_G$ which would uniquely distinguish $G$ among all those with a given property. To this end, we turn to the notion of a signed digraph (a.k.a. "sidigraph"), a digraph where edges have labels $+$ or $-$, as a formalism for making explicit the disjoint set of "positive" and "negative" edges required to uniquely determine $G$. Edges not present in this sidigraph will be the entries we can omit from our adjacency matrix, given those edges which are present and some structural prior.

This framework will allow us to identify the *minimal* set of entries in the adjacency matrix necessary to uniquely distinguish a digraph among all those with a given property. We apply this framework to transitively-closed digraphs, in which case we prove that certain edges can be pruned. We then prove that, in the case of transitively-closed DAGs, this yields the minimal set of entries. We furthermore provide a concrete algorithm to construct this set of entries from a given adjacency matrix.

### 3.1 Preliminary Definitions

While the depiction of the partial adjacency matrix in Figure 2 is clear, in order to formalize this we need a notion of a graph which has three possible values, e.g. 1, 0, and "missing". This is captured formally by the notion of a (simple) **signed digraph**, or **sidigraph** for short, which is a triple $(V, E^+, E^-)$ where $V$ is the vertex set, and $E^+, E^-$ are disjoint subsets of $\text{offdiag}(V)$, referred to as the positive and negative edges, respectively.

**Definition 1.** Given a digraph $G = (V, E)$ we define the **equivalent sidigraph** $G^\pm \coloneqq (V, E, \overline{E})$.

This equivalence defines a bijection between digraphs and sidigraphs with $|V|(|V| - 1)$ edges. See Figure 3 for a depiction of an equivalent sidigraph of the digraph in Figure 1.

Let $H$ and $G$ be two sidigraphs with the same vertex set $V$. We say that $H$ is a **sub-sidigraph** of $G$, denoted by $H \subseteq G$, if $E_H^+ \subseteq E_G^+$ and $E_H^- \subseteq E_G^-$. Given some digraph $G$, identifying a subset of entries in $A_G$ is equivalent to specifying a sub-sidigraph of $G^\pm$. For this reason, we introduce the following terminology.

**Definition 2.** We say that a sidigraph $H$ is a **potential distinguisher** of a digraph $G$ if $H \subseteq G^\pm$. Given a sidigraph $H$, we define the set of digraphs that could **potentially be distinguished** by $H$ to be $\mathcal{G}_H \coloneqq \{G \mid H \subseteq G^\pm\}$.

That is, $\mathcal{G}_H$ contains every digraph $G$ such that $H$ is a potential distinguisher. Another way to interpret $\mathcal{G}_H$ is that the edge set of every graph $G \in \mathcal{G}_H$ contains all positive edges and no negative edges of $H$. More formally: $\mathcal{G}_H = \{G \mid V_G = V_H, E_H^+ \subseteq E_G, E_H^- \cap E_G = \emptyset\}$.

|  | Root | A | B | A1 | A2 | B1 | B2 |
|---|---|---|---|---|---|---|---|
| Root | 0 | 1 | 1 | 1 | 1 | 1 | 1 |
| A | 0 | 0 | 0 | 1 | 1 | 0 | 0 |
| B | 0 | 0 | 0 | 0 | 0 | 1 | 1 |
| A1 | 0 | 0 | 0 | 0 | 0 | 0 | 0 |
| A2 | 0 | 0 | 0 | 0 | 0 | 0 | 0 |
| B1 | 0 | 0 | 0 | 0 | 0 | 0 | 0 |
| B2 | 0 | 0 | 0 | 0 | 0 | 0 | 0 |

(a) Adjacency matrix for $G$

|  | Root | A | B | A1 | A2 | B1 | B2 |
|---|---|---|---|---|---|---|---|
| Root |  | + | + | + | + | + | + |
| A | − |  | − | + | + | − | − |
| B | − | − |  | − | − | + | + |
| A1 | − | − | − |  | − | − | − |
| A2 | − | − | − | − |  | − | − |
| B1 | − | − | − | − | − |  | − |
| B2 | − | − | − | − | − | − |  |

(b) Equivalent sidigraph $G^{\pm}$

|  | Root | A | B | A1 | A2 | B1 | B2 |
|---|---|---|---|---|---|---|---|
| Root |  | + | + |  |  |  |  |
| A |  |  |  | + | + | − | − |
| B |  |  |  | − | − | + | + |
| A1 |  |  |  |  | − |  |  |
| A2 |  |  |  | − |  |  |  |
| B1 |  |  |  |  |  |  | − |
| B2 |  |  |  |  |  | − |  |

(c) Minimal sidigraph $H^*$

Figure 3: An adjacency matrix (a) and a representation of the edges in the associated sidigraph (b), where a $+$ in position $(i, j)$ indicates $(i{\rightarrow}j) \in E^+$ and a $-$ indicates $(i{\rightarrow}j) \in E^-$. The minimal sidigraph (c) formally captures the fact that Figure 2 has the minimal set of entries in the adjacency matrix to uniquely distinguish $G$ among all transitively-closed DAGs.

**Definition 3.** Given some property, we let $\mathcal{P}$ be a set of all digraphs (on a given set of nodes) with this property. We say the sidigraph $H$ **distinguishes** a particular digraph $G$ among all those with the given property if $\mathcal{G}_H \cap \mathcal{P} = \{G\}$.

### 3.2 Distinguishing Transitively-Closed Digraphs

In this work, we focus on the case where $\mathcal{P}$ contains all transitively-closed digraphs, and we let $G = (V, E) \in \mathcal{P}$ be some fixed transitively-closed digraph.

We first focus on reducing the positive edges of $G^{\pm}$. An explicit representation of $G$ may well have $\Omega(|V|^2)$ edges; however any transitive reduction $G' = (V, E')$ often has substantially fewer edges.[3] Furthermore, by definition, $G$ is the only transitively-closed digraph which contains the edges $E'$. As a consequence, if we let $H = (V, E', \overline{E})$ we have $\mathcal{G}_H \cap \mathcal{P} = \{G\}$, which proves the following result.

**Proposition 1.** *Let $G = (V, E)$ be a transitively-closed digraph, and $G' = (V, E')$ a transitive reduction. Then $H = (V, E', \overline{E})$ distinguishes $G$ among transitively-closed digraphs.*

The edge complement $\overline{E}$ might also have $\Omega(|V|^2)$ edges, which we would like to reduce while maintaining distinguishability. To see why this should be possible, consider Figure 4.

If $(a{\rightarrow}b) \in E$ and $(a{\rightarrow}d) \notin E$ then $(b{\rightarrow}d) \notin E$, since if it were we would need to include $(a{\rightarrow}d)$ due to transitivity. Similarly, if $(c{\rightarrow}d) \in E$ and $(a{\rightarrow}d) \notin E$ then $(a{\rightarrow}c) \notin E$, since, if it were, transitivity would imply $(a{\rightarrow}d) \in E$. We formalize this in the following proposition.

**Proposition 2.** *Let $G = (V, E)$ be a transitively-closed digraph; $H = (V, E_H^+, E_H^-)$ a sidigraph which distinguishes $G$ among transitively-closed digraphs. If $(a{\rightarrow}d) \in E_H^-$, then*

*1. If $(a{\rightarrow}b) \in E$, $H' = (V, E_H^+, E_H^- \backslash \{(b{\rightarrow}d)\})$ distinguishes $G$ among transitively-closed digraphs.*

*2. If $(c{\rightarrow}d) \in E$, $H' = (V, E_H^+, E_H^- \backslash \{(a{\rightarrow}c)\})$ distinguishes $G$ among transitively-closed digraphs.*

*Proof.* Recall that $H$ distinguishing $G$ among transitively-closed digraphs means that $\mathcal{G}_H \cap \mathcal{P} = \{G\}$, i.e. $G = (V, E)$ is the only transitively-closed digraph for which $E_H^+ \subseteq E$ and $E_H^- \cap E = \emptyset$. Now, note that $G' = (V, (E_H^+)^{\text{tc}})$ is the smallest transitively-closed digraph containing the edges $E_H^+$, and thus $G' \subseteq G$. But this implies $E_H^- \cap (E_H^+)^{\text{tc}} \subseteq E_H^- \cap E = \emptyset$, and thus $G' = G$.

Now we prove the proposition at hand. We prove the first case, the second follows similarly. Let $(a{\rightarrow}d) \in E_H^-$, assume $(a{\rightarrow}b) \in E$, and define $H' = (V, E_H^+, E_H^- \setminus \{(b{\rightarrow}d)\})$. Note that $H' \subseteq H$, and hence $\mathcal{G}_H \subseteq \mathcal{G}_{H'}$.

Now suppose $K = (V, E_K) \in \mathcal{G}_{H'} \cap \mathcal{P}$, which implies $K$ is a transitively-closed digraph with $E_H^+ \subseteq E_K$ and $(E_H^- \setminus \{(b{\rightarrow}d)\}) \cap E_K = \emptyset$. In particular, as a consequence of our observation in the first paragraph, this means $(a{\rightarrow}b) \in E_K$. We prove $(b{\rightarrow}d) \notin E_K$ by contradiction, for if $(b{\rightarrow}d) \in E_K$, then since $K$ is transitively-closed we would have $(a{\rightarrow}d) \in E_K$ which violates our preliminary assumption. $\qquad \square$

---

[3] A canonical example is the transitive closure of a directed path of length $n$, where $|E^{\text{tc}}| = (n-1)(n-2)$, but $|E^{\text{tr}}| = n-1$.

These two simple prunings, illustrated in Figure 4, lead us to formulate Algorithm 1, FINDMINDIS­TINGUISHER, which repeatedly removes edges using Proposition 2 until no more can be removed.

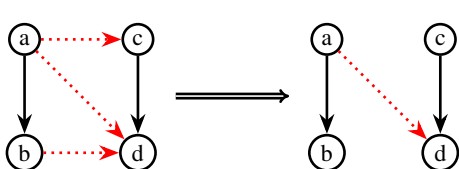

Figure 4: Negative edges removable according to Proposition 2.

---

**Algorithm 1** FINDMINDISTINGUISHER

---

**Require:** $G = (V, E)$ transitively-closed DAG
 1: $E^* \leftarrow \overline{E}$
 2: **for** $(a{\rightarrow}d) \in \overline{E}$ **do**
 3:     **for** $(a{\rightarrow}b) \in E$ **do**
 4:         $E^* \leftarrow E^* \setminus \{(b{\rightarrow}d)\}$
 5:     **end for**
 6:     **for** $(c{\rightarrow}d) \in E$ **do**
 7:         $E^* \leftarrow E^* \setminus \{(a{\rightarrow}c)\}$
 8:     **end for**
 9: **end for**
10: **return** $H^* = (V, E^{\text{tr}}, E^*)$

---

### 3.3 Optimality of FINDMINDISTINGUISHER

We note that $H^* = \text{FINDMINDISTINGUISHER}(G)$ is only defined for DAGs, and indeed for graphs with cycles it is not possible to uniquely define such a sidigraph. (Recall that the transitive closure is not unique for graphs with cycles). In the event that $G$ is acyclic, however, we can prove that $H^*$ is not just capable of distinguishing $G$ among transitively-closed digraphs, but moreover it is the sidigraph with the minimum number of edges capable of doing so! Below is a proof sketch for the informal statement. For the full proof refer to Theorem 1 in Appendix B.

*Proof sketch:* The basic idea is to define a partially ordered set, a.k.a. poset, on the set of negative edges of $G^\pm$. We then show that the set of all minimal elements in this poset is necessary and sufficient to distinguish $G$ among transitively-closed digraphs. Finally, we observe that our algorithm in the previous section produces the set of all minimal elements and hence its optimality follows.

## 4 Leveraging Sufficient Sidigraphs for Training Node Embeddings

We would like to leverage a distinguishing sidigraph for more efficient training and improved accuracy of energy-based node embedding models. The prerequisite is an energy function with a useful inductive bias for the digraph property under consideration. In the extreme case of an inductive bias which only permits models capable of representing digraphs in $\mathcal{P}$ we should be able to train using only the positive and negative edges from any $H$ which can distinguish $G$ among those in $\mathcal{P}$. In some instances, depending on the proportion $[|E_{H^*}^+| + |E_{H^*}^-|]/[|V|(|V| - 1)]$, this may allow full training on digraphs which would otherwise require sampling. As we show, sampling can still be used in conjunction with the minimal edge set implied by a sidigraph, and in such situations we expect not only increased efficiency but increased performance, as $\mathcal{L}_{\text{sampled}}(\theta)$ will better approximate $\mathcal{L}_{\text{full}}(\theta)$.

### 4.1 Transitivity Bias

First, we formally define "transitivity bias".

**Definition 4.** Let $Z$ be an embedding space. We say that an energy function $\mathrm{E}_\theta$ has **transitivity bias** if, for all embeddings $\theta : V \to Z$, there exists some threshold $T \geq 0$ s.t. for all $u, v, w \in V$, the inequalities $\mathrm{E}_\theta(u, v), \mathrm{E}_\theta(v, w) \leq T$ imply $\mathrm{E}_\theta(u, w) \leq T$.

We illustrate transitivity bias using the following overly simple embedding model.

**Example 1** (Bit Vectors). Let $Z = \{0, 1\}^{|V|}$, and $\mathrm{E}_{\text{BV}}(u, v; \theta) := -\log \frac{\theta(u) \cdot \theta(v)}{\theta(v) \cdot \theta(v)}$. Then $\mathrm{E}_{\text{BV}, \theta}$ has a transitivity bias using threshold $T = 0$, as $\mathrm{E}_{\text{BV}}(u, v; \theta) = 0$ if and only if $\forall i \in V$, $\theta(v)_i = 1 \implies \theta(u)_i = 1$. Thus, if $\mathrm{E}_{\text{BV}}(u, v; \theta) = \mathrm{E}_{\text{BV}}(v, w; \theta) = 0$ we have $\theta(w)_i = 1 \implies \theta(v)_i = 1 \implies \theta(u)_i = 1$, hence $\mathrm{E}_{\text{BV}}(u, w; \theta) = 0$.

For a given digraph $G = (V, E)$, we define the bit vector representation of $G$ as $\theta_{\text{BV}} : V \to \{0, 1\}^{|V|}$, where the $i^{\text{th}}$ (for $i \in V$) coordinate is given by

$$\theta_{\mathrm{BV}}(v)_i := \begin{cases} 1 & \text{if } i \text{ equals } v \text{ or is a descendant of } v, \\ 0 & \text{otherwise.} \end{cases}$$

**Proposition 3.** *Let $G = (V, E)$ be any digraph, then $\mathrm{E}_{\mathrm{BV}}(u, v; \theta_{\mathrm{BV}}) = 0$ if and only if $(u, v) \in E^{tc}$.*

*Proof.* Let $(u, v) \in E^{tc}$, then there is some path $u = w_1 \to w_2 \to \cdots \to w_{n-1} \to w_n = v$ in $E$. By the definition of $\theta_{\mathrm{BV}}^G$, we have that $\mathrm{E}_{\mathrm{BV}}(w_i, w_{i+1}; \theta_{\mathrm{BV}}^G) = 0$ for $i \in [\![n]\!]$, and thus by the transitivity bias observation made in Example 1 this implies $\mathrm{E}_{\mathrm{BV}}(u, v; \theta_{\mathrm{BV}}^G) = 0$.

Now assume $(u, v) \notin E^{tc}$, then $v$ is not a descendant of $u$, which means $\theta_{\mathrm{BV}}^G(u)_v = 0$ while $\theta_{\mathrm{BV}}^G(v)_v = 1$, and hence $\mathrm{E}_{\mathrm{BV}}(u, v; \theta_{\mathrm{BV}}^G) > 0$. □

With an appropriate threshold, any energy function that has transitivity bias in fact represents a transitively-closed digraph:

**Proposition 4.** *If $\mathrm{E}_\theta$ is an energy function with transitivity bias, then for any $\theta$ there exists a $T \geq 0$ such that the digraph with edges $\{(u, v) \mid \mathrm{E}_\theta(u, v) \leq T\}$ is transitively closed.*

This allows us to formalize the notion that training on any $H$ which can distinguish $G$ among transitively-closed digraphs is sufficient.

**Proposition 5.** *Let $G$ be a transitively-closed digraph, $\mathrm{E}_\theta$ an energy function with transitivity bias, and $H = (V, E_H^+, E_H^-)$ a sidigraph which distinguishes $G$ among transitively closed digraphs. If $T$ is the threshold associated with the transitivity bias for $\theta$, $\mathrm{E}_\theta(u, v) \leq T$ for all $(u, v) \in E_H^+$, and $\mathrm{E}_\theta(u, v) > T$ for all $(u, v) \in E_H^-$, then $\theta$ represents the digraph $G$.*

## 4.2 Box Embeddings and T-Box

We would like our energy function to be a representation which is tractable and trainable via gradient-descent, which requires the space $Z$ to have differentiable structure. *Box embeddings* [Vilnis et al., 2018] are a trainable region-based embedding method which demonstrate strong performance for representing digraphs [Boratko et al., 2021a, Zhang et al., 2022]. We provide the requisite background on box embeddings and define the specific model that we will use for our experiments.

As introduced in Vilnis et al. [2018], box embeddings represent entities using a *box* or *hyperrectangle* in $\mathbb{R}^D$, *i.e.*, a Cartesian product of intervals

$$\prod_{d=1}^{D} [x_d^\llcorner, x_d^\urcorner] = [x_1^\llcorner, x_1^\urcorner] \times \ldots \times [x_D^\llcorner, x_D^\urcorner] \subseteq \mathbb{R}^D,$$

where $x_d^\llcorner < x_d^\urcorner$ for $d \in [\![D]\!]$. Vilnis et al. [2018] proposed modeling a directed graph such that boxes of parents contain their children with an energy function

$$\mathrm{E}_{\mathrm{Box}}(u, v; \theta) := -\log \prod_{d=1}^{D} F_{\mathrm{Box}}(\theta(u)_d, \theta(v)_d),$$

where the per-dimension parameters are endpoints of an interval, *i.e.* $\theta(u)_d = [\theta(u)_d^\llcorner, \theta(u)_d^\urcorner]$, and the per-dimension score is defined as

$$F_{\mathrm{Box}}((x^\llcorner, x^\urcorner), (y^\llcorner, y^\urcorner)) := \frac{|[x^\llcorner, x^\urcorner] \cap [y^\llcorner, y^\urcorner]|}{|[y^\llcorner, y^\urcorner]|} = \frac{\max(\min(x^\urcorner, y^\urcorner) - \max(x^\llcorner, y^\llcorner), 0)}{\max(y^\urcorner - y^\llcorner, 0)}.$$

Previous works have highlighted the difficulty of optimizing an objective including these hard $\min$ and $\max$ functions [Li et al., 2019, Dasgupta et al., 2020]. We use the Global T-Box model, the most recent solution to this issue, introduced in Boratko et al. [2021a]. Global T-Box (or GT-Box) softens the volume calculation by replacing the hard $\min$ and $\max$ operators with a smooth approximation $\mathrm{LSE}_t(\mathbf{x}) := t \log(\sum_i e^{x_i/t})$. The per-dimension score function is then given by

$$F_{\mathrm{GT\text{-}Box}}((x^\llcorner, x^\urcorner), (y^\llcorner, y^\urcorner); \tau, \nu) := \frac{\mathrm{LSE}_\nu(\mathrm{LSE}_{-\tau}(x^\urcorner, y^\urcorner) - \mathrm{LSE}_\tau(x^\llcorner, y^\llcorner), 0)}{\mathrm{LSE}_\nu(y^\urcorner - y^\llcorner, 0)},$$

which approximates $F_{\mathrm{Box}}$ for sufficiently small $\tau, \nu > 0$. These $\tau, \nu$ are additional (global) trainable parameters of the model.

### 4.3 Transitivity Bias of Box Embeddings

While the motivation and formulation of the energy for box embedding functions is quite different than that of the naive bit vector model in Example 1, it actually is more similar than it may at first appear.

**Remark 1.** As observed in Boratko et al. [2021b], the box intersection volume calculation can actually be viewed as the $L^2$ inner-product of characteristic functions of boxes, in which case it takes an identical form to the energy function for bit vectors. For any functions $f, g \in L^2(\mathbb{R}^D)$, the standard inner product is $\langle f, g \rangle = \int_{\mathbb{R}^D} f(x)g(x)\,dx$. Now, for any box $B \subseteq \mathbb{R}^D$, we let $\mathbb{1}_B(x)$ be the characteristic function, which is 1 when $x \in B$ and 0 otherwise. Then the volume of intersection of boxes is

$$\mathrm{Vol}(A \cap B) = \langle \mathbb{1}_A(x), \mathbb{1}_B(x) \rangle,$$

which is also valid if $A = B$. Thus, we can write the energy function

$$\mathrm{E}_{\mathrm{Box}}(u, v; \theta) = -\log \frac{\langle \mathbb{1}_{\theta(u)}, \mathbb{1}_{\theta(v)} \rangle}{\langle \mathbb{1}_{\theta(v)}, \mathbb{1}_{\theta(v)} \rangle}.$$

**Remark 2.** Box embeddings can be quantized into bit vectors in such a way that applying the bit vector energy function to the resulting quantizations preserves the set of node pairs which have zero energy. Given a box embedding $\theta$, for each $d \in [\![D]\!]$ let $T_d$ be the endpoints of boxes in dimensions $d$, *i.e.* $T_d := \cup_{v \in \mathcal{V}} \{\theta(v)_d^-, \theta(v)_d^+\}$. Let $M_d := |T_d|$, and assign indices $T_d =: \{t_{d,m}\}_{m=1}^{M_d}$ such that $t_{d,m} \leq t_{d,m+1}$ for $m \in [\![M_d - 1]\!]$. Then, for each $v \in \mathcal{V}$ and $d \in [\![D]\!]$, form the $M_d$-dimensional vector $\varphi(v)_{d,m} := F_{\mathrm{Box}}(\theta(v)_d, (t_{d,m}, t_{d,m+1}))$. By construction, this value will be either 0 or 1. Letting $\varphi(v) \in \{0, 1\}^{\sum M_d}$ be the concatenation of $\{\varphi(v)_d\}_{d=1}^D$, we obtain a bit vector representation, for which the energy function $E(u, v; \varphi) = -\log \frac{\varphi(u) \cdot \varphi(v)}{\varphi(v) \cdot \varphi(v)}$ is such that $E(u, v; \varphi) = 0$ if and only if $\varphi(v)_{d,m} = 1 \implies \varphi(u)_{d,m} = 1$, which is true if and only if $[\theta(v)_d^+, \theta(v)_d^+] \subseteq [\theta(u)_d^-, \theta(u)_d^+]$.

Most importantly for our purposes, however, is the following proposition.

**Proposition 6.** *The energy function $\mathrm{E}_{\mathrm{Box}, \theta}$ has a transitivity bias.*

Apart from prior empirical observations that box embeddings work well to embed transitively-closed DAGs Boratko et al. [2021a], Proposition 5 suggests it should be possible to train box embeddings on the output of FINDMINDISTINGUISHER to represent a transitively-closed DAG. In practice, we train the smooth approximation provided by GT-Box. As $\tau, \nu \to 0^+$, which often happens naturally during training Boratko et al. [2021a], we expect it to capture the transitivity even when trained only on the positive and negative edges of the distinguisher provided by the algorithm.

## 5 Hierarchy-Aware Sampling

In this section we formally how we sample edges for our loss function, which we term hierarchy-aware sampling. Based on the results in the preceding section, given a transitively-closed digraph $G$, if $\mathrm{E}_\theta$ has a transitive bias and the sidigraph $H = (V, E_H^-, E_H^+)$ can distinguish $G$ among transitively-closed digraphs, then we can train using the following loss function:

$$\mathcal{L}_{\mathrm{ha}}(\theta; H) := \sum_{(u \to v) \in E_H^+} \ell^+(\mathrm{E}_\theta(u, v)) + \sum_{(u \to v) \in E_H^-} \ell^-(\mathrm{E}_\theta(u, v)).$$

In particular, for a transitively-closed DAG $G$, we can use $H^*$ as returned by Algorithm 1.

If the cardinality of $H^*$ is small enough, this may make training using $\mathcal{L}_{\mathrm{ha}}$ feasible. In general, however, we still may need to sample negative edges using some noise distribution, as mentioned in Section 2.2. In practice, we will compare using the sidigraphs $G^\pm$ and $H_* = $ FINDMINDISTINGUISHER$(G)$, which means positives will be sampled from either $E$ or $E^{\mathrm{tr}}$, and negatives will be sampled from $\overline{E}$ or $E_{H^*}^-$. Even with our reduced set of negatives, there are still far more negatives than positives, and so we adopt the common practice of sampling $k$ negatives for every positive within a batch.

# 6 Experiments

The main aim of our experiments is to compare the efficacy on graph representation learning of models with vs without transitivity bias, and random uniform vs hierarchy-aware sampling. As per Proposition 5, we hypothesize that the greatly reduced edge set in hierarchy-aware sampling will be sufficient to faithfully represent the graphs in the embedding space if our model has transitivity bias.

**Data.** We evaluate hierarchy-aware sampling on the heterogeneous synthetic DAGs for which Boratko et al. [2021a] demonstrated superior performance using GT-BOX and random uniform negative sampling: **balanced trees**, where $b$ is the branching factor, the **nested Chinese restaurant process (nCRP)** [Blei et al., 2010], where $\alpha$ is the normalized "new table" probability, and **Price's model** [Price, 1976], where $m$ is the number of connections for a new node and $c$ is a constant factor added to the probability of a vertex receiving an edge[4]. We also test on the larger real-world Medical Subject Headings (MeSH) taxonomy Lipscomb [2000], 2020 release. For detailed statistics about all graphs we refer the reader to Table 1 in Appendix F.

**Models.** To evaluate the importance of the model having transitivity bias in order to take advantage of this reduced set of edges, we consider the vector similarity model SIM-VEC which represents each node by an "in" and "out" vector, and computes edges via dot-product, $E_{\text{SIM-VEC}}(u, v) := -\log(\sigma(\theta(u)_{\text{out}} \cdot \theta(v)_{\text{in}}))$. Unlike GT-BOX, SIM-VEC does not have transitivity bias Boratko et al. [2021a]. For all of our experiments we fix the dimension to $64$; this corresponds to $64$-dimensional "in" and "out" embeddings for SIM-VEC, and to the two $64$-dimensional corners $x^{\llcorner}$ and $x^{\urcorner}$ of GT-BOX. This makes the number of parameters per node for both models 128. For details on hyperparameter tuning, see Appendix E.

**Support and Sampling.** For the synthetic graphs we iterate over settings relating to the support sets from which to uniformly randomly sample positive and negative training edges, respectively:

- Sampling positive edges from the transitive reduction $E^{\text{tr}}$ vs transitive closure $E^{\text{tc}}$.
- Sampling negative edges from minimal hierarchy-aware set $E_{H^*}^-$ vs edge complement $\overline{E}$.
- Negative sampling ratio $k = 4$ vs $k = 128$ (i.e. per each positive edge seen by the model, $k$ negative edge sampled uniformly randomly from the support set also get seen).

Setting $k = 4$ mimics the tiny proportion of sampled negatives to the pool of all possible negatives, a limitation which we expect when scaling to larger graphs, such as MeSH.

**Evaluation.** Since we are evaluating the representational capacity of each experimental setting (as opposed to generalization on some held-out edge set), our metric is the F1 between the edges of the transitively-closed DAG we are modeling and the model's scores for those edges. In other words, we are performing binary classification over the full adjacency matrix for the hierarchy in question.

We investigate the impact of the available positive and negative support sets on model training; since the representational capacities of vector and box models are well-studied, we are *not* interested in the best F1 score attainable by these models, but the effect of respective support sets on convergence. To compare across experimental settings, we plot F1 as a function of the number of total examples (both positive and negative edges) processed by the model (e.g. Figure 6). To measure the final performance and rate of convergence, we use the area under the F1 vs. Total Examples curve (AUF1C).

## 6.1 Results and Discussion

First note that GT-BOX universally outperforms SIM-VEC on the graph modeling experiments. This is not surprising, as Boratko et al. [2021a] demonstrate the superior inductive bias of GT-BOX for modeling digraphs generally. The aim of our experiments, however, is to investigate the impact of a hierarchy's structure on the convergence for the various sampling settings for GT-BOX specifically. With this aim, in Figure 5, we present the AUF1C for the convergence curve (area under F1 vs. number of examples) in the limited-example setting of $k = 4$. In this regime random uniform negative sampling may not be able iterate over all of $\overline{E}$ in a reasonable amount of time, whereas, to the contrary, it is possible to get through all of $E_{H^*}^-$ in a limited number of epochs, since in many cases the sampling pool is reduced by more than 99% (cf. Table 1).

---

[4]Preferential attachment exponent $\gamma = 1.0$ throughout.

| Model | Training Edges | | Graph Type | | |
|---|---|---|---|---|---|
| | Pos. | Neg. | BT | nCRP | Price |
| GT-Box | ◐ | ◐ | 0.568 | 0.796 | 0.582 |
| | ◐ | ● | 0.658 | 0.775 | 0.602 |
| | ● | ◐ | **0.842** | **0.904** | 0.849 |
| | ● | ● | 0.744 | 0.877 | **0.909** |
| SIM-VEC | ◐ | ◐ | 0.072 | 0.084 | 0.158 |
| | ◐ | ● | 0.180 | 0.285 | 0.336 |
| | ● | ◐ | 0.098 | 0.158 | 0.300 |
| | ● | ● | 0.508 | 0.409 | 0.494 |

Figure 5: Convergence measured using AUF1C on three graph types for SIM-VEC and GT-BOX models with the full or reduced set of positive and negative edges provided during training, respectively. Pos. = ● (resp. ◐) implies usage of transitive closure (resp. transitive reduction). Analogously, for Neg., the symbols imply usage of the set of the edge complement $\overline{E}$ and $E_{H^*}^-$, respectively.

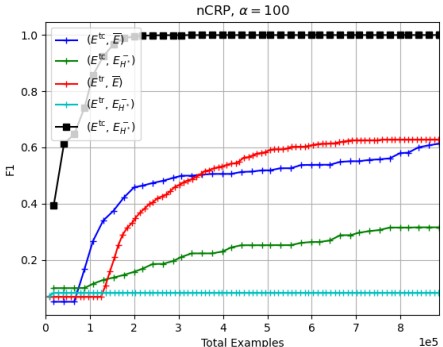

Figure 6: While GT-BOX (black squares) with a transitivity bias takes full advantage of the hierarchy-aware $E_{H^*}^-$, SIM-VEC, which does not have a transitivity bias, falls apart under this negative sampling procedure. The reduction in negative examples from using the pruned negative edges for this particular graph is $95.83\%$. Negative ratio $k = 4$ throughout.

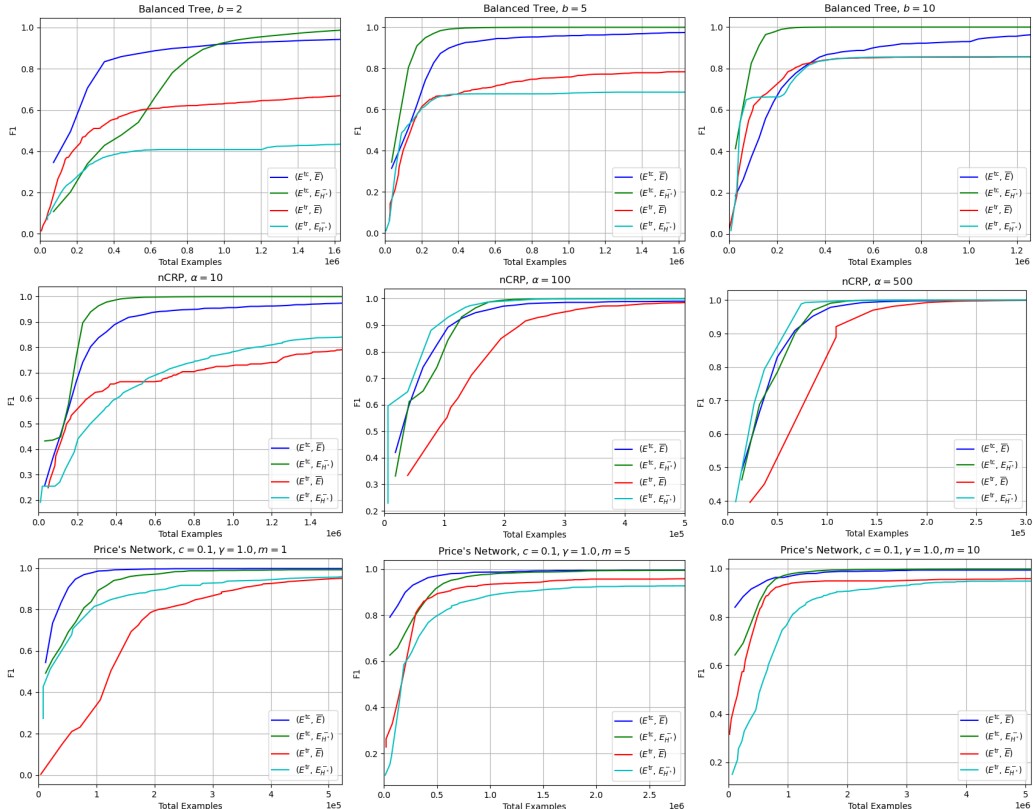

Figure 7: The plots show convergence of GT-BOX for negative ratio $k = 4$. The top row shows the plots for balanced trees with branching factors $b = 2, 5$ and $10$. The middle row for nCRP graphs with $\alpha = 10, 100$, and $500$, respectively going left to right. The bottom row shows the plots for Price's graph with $c = 0.1$, $\gamma = 1.0$ and three values of $m = 1, 5$, and $10$. The number of vertices in each graph is $\approx 2^{13}$.

Figure 5 demonstrates a striking trend observed consistently in our experiments. **While GT-BOX with our reduced edge set performs on par or better than with the original edge set, in contrast, the performance of SIM-VEC degrades significantly.** We demonstrate this using a representative example in Figure 6, where, in stark contrast to GT-BOX taking advantage of $E_{H^*}^-$ and outperforming all SIM-VEC settings, both SIM-VEC settings that use $E_{H^*}^-$ fail to converge even to 0.4 F1. This dichotomy hearkens back to Proposition 5, which underlines the non-trivial synergy between an energy function with transitivity bias and the hierarchy-aware output of FINDMINDISTINGUISHER.

**Balanced tree.** Figure 11 Row 1 visualizes the impact on convergence of increasing the branching factor. As branching factor $b$ increases $2 < 5 < 10$, we first note that the balanced tree with fixed $|V|$ becomes more and more shallow. While $E_{H^*}^-$ performs well on all values of $b$, it particularly stands apart from $\overline{E}$ in the $b = 10$ setting.

**nCRP.** $\alpha$ is the normalized "new table" probability, with a smaller $\alpha$ implying more separate clusters with a deeper hierarchy, and a larger $\alpha$ implying fewer clusters and a more shallow hierarchy. The relative improvement of using $E_{H^*}^-$ is more significant for smaller $\alpha$.

**Price.** Price's model is learnable by GT-BOX especially quickly under any setting, as evidenced by the very high AUF1C scores in Figure 5. Higher $c$ has the effect of making edge attachment more uniformly distributed among nodes. In Figure 11 Row 3 we examine convergence values for $c = 0.1$ , noting that $E_{H^*}^-$ typically underperforms $\overline{E}$ but catches up in the limit. As $m$ (out-degree of newly added vertices) increases $1 < 5 < 10$, looking at Table 1 we note that hierarchy-aware negative pruning also gets more aggressive as $52.35\% < 89.62\% < 93.77\%$.

**MeSH.** As seen in Figure 8, on the larger real-world MeSH taxonomy, not only does GT-BOX outperform SIM-VEC, but our minimal negative edge set $E_{H^*}$ within GT-BOX outperforms sampling from the edge complement, while being a 99.78% reduction. Meanwhile, $E_{H^*}$ with SIM-VEC plummets, consistently with the trend exhibited in Figure 6. This encouraging result on a graph with $\approx 2^{15}$ nodes points not only to the economy of our approach, but to the stability and improved performance on large real-world data.

# 7 Limitations

While the connection between hierarchies and transitivity bias allows us to capitalize on it in the form of hierarchy-aware sampling, we acknowledge that the properties demanded of both the data and model are restricted to transitivity. While this does encompass a large variety of relationships observed in real-world graphs, this specific algorithm does not extend easily to new combinations of graph properties and inductive biases, which is a goal of future work.

Another limitation is the extent to which this sort of method would break down for graphs which are, strictly speaking, not transitively-closed, but close (in edit distance) to being transitively-closed. Strictly speaking, our proofs do not apply in that setting, and the efficacy of the approach may vary depending on the type of structural changes a particular removal of an edge brings about.

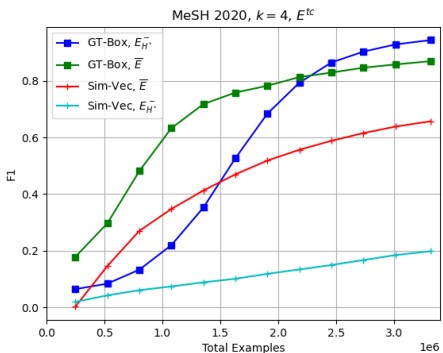

Figure 8: Convergence of GT-BOX vs SIM-VEC for $\overline{E}$ vs $E_{H^*}$ on MeSH 2020. The minimal negative edge set $E_{H^*}$ converges to the highest F1 when coupled with GT-BOX, but falls apart when combined with SIM-VEC, consistently with our hypothesis about the requirement of transitivity bias for utilizing $E_{H^*}$.

# 8 Conclusion

In this work, we propose a novel framework for identifying a sufficient subset of entities in the adjacency matrix which unambiguously specify a digraph, given some prior structural knowledge. We demonstrate the usability of this framework for the property of transitively-closed DAGs, or hierarchies. We derive a characterization of the sufficient negative set for such type of graphs, and based on that we devise a novel hierarchy-aware sampling technique. Our approach is efficient and robust when used in conjunction with an energy-based node embedding model possessing the appropriate inductive bias, which, for hierarchies, is transitivity bias.

# 9 Acknowledgements

This work was supported in part by IBM Research AI through the AI Horizons Network, the Chan Zuckerberg initiative under the project Scientific Knowledge Base Construction, and National Science Foundation (NSF) grant number IIS-1922090. Hung Le was supported by an NSF CAREER Award No. CCF-223728, an NSF Small Grant No. CCF-2121952, and a Google Research Scholar Award.

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

# A  Related Work

## A.1  Negative Sampling for Graph Representation Learning

Yang et al. [2020] conceptually unify much of the research on negative sampling (NS) for graph representation learning (GRL) into the *SampledNCE* framework which is grounded in noise-contrastive estimation [Gutmann and Hyvärinen, 2012, Mnih and Teh, 2012]. Additionally, they demonstrate that (for vector node representations), the optimal negative sampling distribution is positively but sub-linearly correlated to the positive sampling distribution, *i.e.* $p_n(u \mid v) \propto p_d(u \mid v)^\alpha, 0 < \alpha < 1$, which supports the empirically-determined $3/4$ in Mikolov et al. [2013]'s degree-based sampling.

Examples of better-than-uniform NS for GRL include, in chronological order, Markov chain Monte Carlo NS (MCNS) [Yang et al., 2020], self-adversarial NS (SANS) [Kamigaito and Hayashi, 2022], Adaptive NS (AdaNS) [Wang et al., 2023]. A handful of approaches leverage the graph structure directly to gather negative examples; in particular, Structure Aware NS (also abbreviated SANS) of Ahrabian et al. [2020] selects negatives from a node's $k$-hop neighborhood. Hierarchical Negative Sampling (HNS) [Chen et al., 2021] is only close to our HANS algorithm in nomenclature, relying on a hierarchical dirichlet process to model neighbor proximity information, while borrowing its approach from DeepWalk's [Perozzi et al., 2014] random walks for skip-gram prediction of the current vertex.

We emphasize that each of these above approaches presupposes undirected graphs and vector embeddings of nodes, aligning them with the vast literature on KG embeddings including TransE [Bordes et al., 2013], TransR [Lin et al., 2015], RESCAL [Nickel et al., 2011], DistMult [Yang et al., 2015], ComplEx [Trouillon et al., 2016], RotatE [Sun et al., 2019a], and Rot-Pro [Song et al., 2021] (the last of which models transitive relations via a projection operator).

## A.2  Hierarchical Representations

Box embeddings can represent any DAG [Boratko et al., 2021a], and more recently have been extended to represent any digraph [Zhang et al., 2022]. Empirically, box embeddings have demonstrated particularly strong performance on transitively-closed graphs, making them a strong candidate for modeling hierarchies [Patel et al., 2020]. The use of regions to capture edges in a graph has roots in classic graph invariants of *boxicity* [Roberts, 1969] and *sphericity* [Maehara, 1984], which are defined in the undirected setting. In general, one might expect region-based embeddings to have a natural bias toward modeling transitivity, as the containment relation is transitive [Patel et al., 2022]. Additional region-based embeddings which use cones [Vendrov et al., 2016, Lai and Hockenmaier, 2017] or discs [Suzuki et al., 2019] have also been proposed.

Another prominent line of work for graph representation leverages the negative curvature of hyperbolic space to embed trees without distortion [Sarkar, 2011, Weber and Nickel, 2018, Weber, 2020]. A variety of approaches to training hyperbolic representations with gradient descent have been proposed, to great success [Nickel et al., 2015, Law et al., 2019]. The highlight of these methods is their ability to represent trees, but they are not necessarily well-aligned with representing transitively-closed digraphs [Boratko et al., 2021a]. Hyperbolic entailment cones [Ganea et al., 2018] combine the benefits of both hyperbolic space and region-based containment transitivity.

## A.3  Learning Node Embeddings

The idea of optimizing a loss which is a sum over positive and negative edges captures the approach of training node embeddings in all the work mentioned in the preceding section, however, there are also a variety of other approaches to learning node embeddings. DeepWalk [Perozzi et al., 2014] and node2vec [Grover and Leskovec, 2016] use random walk-based algorithms to generate node embeddings for undirected graphs. LINE [Tang et al., 2015] was the first attempt to address the scalability of learning node embeddings for large-scale graphs. Meanwhile, some works focus on learning node embeddings for directed graphs [Ou et al., 2016, Sun et al., 2019b, Sim et al., 2021]. While more conventionally used in settings where node features are present, graph neural networks (GNNs) [Kipf and Welling, 2016], which iteratively aggregate node features using the local graph structure, have been used to learn node embeddings [Chen et al., 2020, Liu et al., 2020, 2021]. While these approaches are not directly addressed in this work, the general notion of using a sufficient sidigraph to reduce the set of edges and non-edges could be leveraged in such contexts as well.

## A.4 Uniquely Distinguishing Graphs

The idea of uniquely distinguishing a graph with respect to a property has been explored in graph theory literature, albeit without a special focus on DAGs as in the current work. For a property $P$ on $n$-vertex graphs, Adamaszek [2014] defines $P$ to be "nonevasive" (or "nonelusive") in the context of a 2-player game where for each turn, player A is only allowed to ask player B whether a pair of nodes forms an edge or not. $P$ is nonevasive if it can be determined by A in strictly fewer than $\binom{n}{2}$ turns, *i.e.* by asking fewer questions than edges in a complete graph over $n$ vertices (it follows that completeness is an evasive property). Kelenc et al. [2018] spotlight minimal uniquely distinguishing graphs by introducing the concept of an edge metric dimension as the smallest cardinality of a set $S$ such that every pair of edges in $G$ is distinguished (w.r.t. node-edge distance) by some vertex in $S$. An appealing application of this graph-theoretic line of thought might be selecting the smallest set of edges from an ontology for the purpose of human annotations to facilitate active learning; Peng et al. [2018] present an algorithm for enumerating consistent subgraphs of DAGs, which are often used for ontologies in the biomedical domain.

# B   Proof of Optimality

For notational convenience, for each vertex $a \in G$, we add a directed self-loop $(a{\rightarrow}a)$ to indicate that $a$ is reachable from itself. This self-loop will be a positive edge in $G^{\pm}$. Adding this self-loop does not destroy the property that $G$ is transitively closed.

## B.1   A poset for negative edges

We define an order $\preceq$ between negative edges as follows.

For every negative edge $(a{\rightarrow}b)$, we impose $(a{\rightarrow}b) \preceq (a{\rightarrow}b)$. Thus, $\preceq$ is reflexive. Next, we define an order between two negative edges outgoing from or incoming to the same vertex. Let $a$ be an arbitrary vertex in $V$. Let $(a{\rightarrow}b)$ and $(a{\rightarrow}c)$ be two *negative edges out-going from* $a$. Then we define $(a{\rightarrow}b) \preceq (a{\rightarrow}c)$ if $(c{\rightarrow}b)$ is a *positive* edge in $G^{\pm}$ (in other words, if $(c{\rightarrow}b) \in E$). We call the edge pair $((a{\rightarrow}b), (a{\rightarrow}c))$ a **primitive pair**. See Figure 9(a). Similarly, let $(x{\rightarrow}a)$ and $(y{\rightarrow}a)$ be any two *negative edges incoming to* $a$. We define $(x{\rightarrow}a) \preceq (y{\rightarrow}a)$ if $(x \rightarrow y)$ is a positive edge. The edge pair $(x{\rightarrow}a)$ and $(y{\rightarrow}a)$ is also a primitive pair.
Finally, we extend $\preceq$ to satisfy transitivity: if there exist three edges such that $(a{\rightarrow}b) \preceq (c{\rightarrow}d)$ and $(c{\rightarrow}d) \preceq (e{\rightarrow}f)$ such that currently there is no relationship between $(a{\rightarrow}b)$ and $(e{\rightarrow}f)$ under $\preceq$, then we impose $(a{\rightarrow}b) \preceq (e{\rightarrow}f)$.

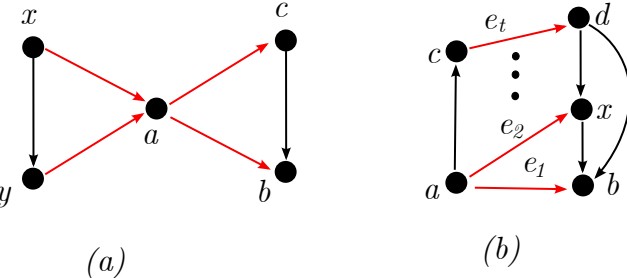

(a)                                                        (b)

Figure 9: (a) Two primitive pairs $((a{\rightarrow}b), (a{\rightarrow}c))$ and $((x{\rightarrow}a), (y{\rightarrow}a))$. (b) Illustrating the proof of Lemma 1.

Our goal is to show that $(\overline{E}, \preceq)$ is a poset. To this end, we characterize all pairs of negative edges $(a{\rightarrow}b)$ and $(c{\rightarrow}d)$ such that $(a{\rightarrow}b) \preceq (c{\rightarrow}d)$.

**Lemma 1.** *Let $(a{\rightarrow}b)$ and $(c{\rightarrow}d)$ be two edges such that $(a{\rightarrow}b) \preceq (c{\rightarrow}d)$. (It could be that $a = c$ and/or $b = d$.) Then $(a \rightarrow c)$ and $(d \rightarrow b)$ are positive edges.*

*Proof.* We prove by induction. The base cases are for primitive pairs. If $(a{\rightarrow}b)$ and $(c{\rightarrow}d)$ are a primitive pair, then either $a = c$ or $b = d$. In both cases, the lemma holds by definition of the order $\preceq$ on primitive pairs.

Let $(a{\rightarrow}b) = e_1 \preceq e_2 \preceq \ldots \preceq e_t = (c{\rightarrow}d)$ be a minimal sequence of orders realizing the order $(a{\rightarrow}b) \preceq (c{\rightarrow}d)$ such that every consecutive pair of negative edges $(e_i, e_{i+1})$ for every $i \in [1, t-1]$ is a primitive pair. Specifically, $(e_1, e_2)$ is a primitive pair. We consider two cases: (i) $e_2 = (a{\rightarrow}x)$ and (ii) $e_2 = (y{\rightarrow}b)$ for some vertex $y$. The two cases are symmetric, so we only focus on the first case; see Figure 9(b). As $(a{\rightarrow}b) \preceq (a{\rightarrow}x)$, $(x \rightarrow b)$ is a positive edge by the definition of $\preceq$ on primitive pairs. Since $(a{\rightarrow}x) \preceq (c{\rightarrow}d)$, by induction, $(a \rightarrow c)$ and $(d \rightarrow x)$ are positive edges. This means $(d \rightarrow b)$ is also a positive edge since $G$ is transitively closed. Thus, the lemma holds.   □

We know that Lemma 1 holds even when $(a{\rightarrow}b) = (c{\rightarrow}d)$ as in that case $a = c$ and $b = d$ and both self-loops $(a \rightarrow a)$ and $(b \rightarrow b)$ are positive. Now, we are ready to show that $\preceq$ is anti-symmetric; the proof uses the fact that $G$ is a DAG.

**Lemma 2.** *If $G = (V, E)$ is a transitively-closed DAG, then $(\overline{E}, \preceq)$ is a poset.*

*Proof.* The order $\preceq$ is transitive and reflexive by definition. We now show anti-symmetric by contradiction. Suppose that $(a{\rightarrow}b)$ and $(x{\rightarrow}y)$ are two *different negative edges* such that $(a{\rightarrow}b) \preceq$

$(x{\rightarrow}y)$ and $(x{\rightarrow}y) \preceq (a{\rightarrow}b)$. Since $(a{\rightarrow}b) \neq (x{\rightarrow}y)$, either $x \neq a$ or $b \neq y$. We focus on the case $x \neq a$; the proof for the case $b \neq y$ is the same.

Since $(a{\rightarrow}b) \preceq (x{\rightarrow}y)$, by Lemma 1, $(a \rightarrow x)$ is a positive edge. Also by Lemma 1, as $(x{\rightarrow}y) \preceq (a{\rightarrow}b)$, we have that $(x \rightarrow a)$ is also a positive edge. However, this contradicts that $G$ is a DAG. $\square$

We could reinterpret Proposition 2 in the language of the poset $(\overline{E}, \preceq)$:

**Corollary 1.** *Let $G$ be a transitively closed DAG. Let $H = (V, E_H^+, E_H^-)$ be a sidigraph which distinguishes $G$ among transitively-closed digraphs. Let $e_1, e_2$ be a primitive pair of negative edges in $E_H^-$ such that $e_1 \preceq e_2$. Then $(V, E_H^+, E_H^- \setminus \{e_2\})$ also distinguishes $G$ among transitively-closed digraphs.*

### B.2 A proof of optimality

We have shown in Lemma 2 that $(\overline{E}, \preceq)$ is a poset. Let $M_\preceq$ be the set of minimal elements of this poset; see Figure 10. Our proof to show that Algorithm 1 is optimal has two steps: (Step 1) the negative edges in $M_\preceq$ are necessary and sufficient for any sidigraph which distinguishes $G$ among transitively-closed digraphs, and (Step 2) the sidigraph $H$ output by our algorithm has $E_H^- = M_\preceq$.

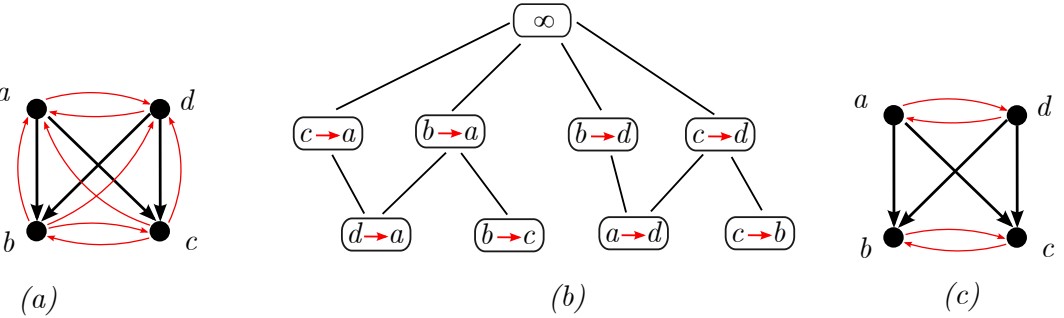

Figure 10: (a) An equivalent sidigraph $G^\pm$. (b) The Hasse diagram of the poset on the negative edges with an artificially added element $\infty$. Here $M_\preceq = \{(d{\rightarrow}a), (b{\rightarrow}c), (a{\rightarrow}d), (c{\rightarrow}b)\}$ (c) a minimal sidigraph $(V, E^{\mathrm{tr}}, M_\preceq)$ which can distinguish $G$ among transitively-closed digraphs.

**Proposition 7.** *If $G = (V, E)$ is a DAG, then $H = (V, E^{tr}, M_\preceq)$ can distinguish $G$ among all transitively-closed digraphs.*

*Proof.* Let $L$ be the linear ordering of all edges in $\overline{E}$ in non-decreasing order of $\preceq$. That is, if $(x{\rightarrow}y) \preceq (a{\rightarrow}b)$, then $(a{\rightarrow}b)$ appears before $(x{\rightarrow}y)$ in $L$. $L$ can be obtained by visiting the negative edges in the BFS order in the Hasse diagram with an artificially added maximum element $\infty$ to define the root of the BFS tree; see Figure 10(b).

Let $X_0 = \overline{E}$ and $i = 0$. We then visit all negative edges by the order in $L$. For each negative edge $e$ visited in this order, if $e \notin M_\preceq$, we define $X_{i+1} = X_i \setminus \{e\}$ and increase $i$ by 1. Let $X_t$ be the last set of negative edges after this process terminates. Clearly, $X_t = M_\preceq$.

By Proposition 1, $(V, E^{tr}, X_0)$ can distinguish $G$. Inductively, we assume that $(V, E^{tr}, X_i)$ can distinguish $G$ for some $i$. Let $e$ be a negative edge such that $X_{i+1} = X_i \setminus \{e\}$. Since $e \notin M_\preceq$, there exists a primitive pair of negative edges $e' \preceq e$. Since $e'$ appears after $e$ in $L$, both edges are present in $X_i$. By Corollary 1, $(V, E^{tr}, X_{i+1})$ can distinguish $G$ among transitively closed digraphs.

We have shown by induction that $(V, E^{tr}, X_t)$, which also is $(V, E^{tr}, M_\preceq)$, is a transitively-closed distinguisher. $\square$

Next, we show that any sidigraph capable of distinguishing $G$ among transitively-closed digraphs must contain any $M_\preceq$ in it's negative set.

**Proposition 8.** *Let $H = (V, E_H^-, E_H^+)$ be any sidigraph capable of distinguishing $G$ among transitively-closed digrphs. Then $M_\preceq \subseteq E_H^-$.*

*Proof.* We prove by contraction: suppose $e = (c{\to}d)$ is such that $(c{\to}d) \in M_\preceq \setminus E_H^-$. Let $G' = (V, E \cup \{e\})$. We show below that $G' \in \mathcal{G}_H \cap \mathcal{P}$, which then means that $H$ cannot distinguish $G$ from $G'$ among transitively-closed digraphs.

Note that $G$ contains all positive edges of $H$ since $H$ is a transitively-closed distinguisher. Thus, $G'$ contains all positive edges of $H$. Next, we show that $G'$ does not contain any negative edge. Suppose otherwise, $G'$ contains a negative $(a{\to}b) \in E_H^-$. Since $(a{\to}b) \notin E$, the only way for $(a{\to}b) \in E \cup \{e\}$ is because of $(c{\to}d)$, and in particular, $(a{\to}c)$ and $(d{\to}b)$ are both positive edges in $G$. Thus, by Lemma 1, $(a{\to}b) \preceq (c{\to}d)$, contradicting that $(c{\to}d) \in M_\preceq$.

Finally, we show that $G'$ is transitively-closed. Suppose otherwise, there are two vertices $a$ and $b$ such that $b$ is reachable from $a$ but $(a \to b) \notin G'$. Note that $b$ is not reachable from $a$ in $G$ since $G$ is transitively closed. This implies: (i) the path from $a$ to $b$ in $G'$ must contain the edge $(c{\to}d)$ and (ii) $(a{\to}b)$ is a negative edge in $G$. By (i), it must be that $(a{\to}c)$ and $(d{\to}b)$ are positive edges. By (ii) and Lemma 1, $(a{\to}b) \preceq (c{\to}d)$, contradicting that $(c{\to}d) \in M_\preceq$. $\qquad\square$

Proposition 7 and Proposition 8 together imply that any transitively closed-distinguisher of $G$ with $M_\preceq$ contains a minimum number of negative edges. Thus, to show that our algorithm gives a transitively closed distinguisher of $G$, it suffices to show that the set of negative edges is $M_\preceq$.

**Theorem 1.** *Let $G = (V, E)$ be any DAG, and let $H$ be the output of* FINDMINDISTINGUISHER$(G)$, *so $H = (V, E^{tr}, M_\preceq)$. Then $H$ can distinguish $G$ among transitively closed digraphs, and has the minimal number of edges of any such sidigraph.*

*Proof.* We focus on showing that $E_H^- = M_\preceq$. The latter claim follows from Proposition 7 and Proposition 8.

Observe that the algorithm considers primitive pairs and, for each pair, removes the negative edge with a higher order in the poset $(\overline{E}, \preceq)$. Thus, the algorithm never removes a negative edge in $M_\preceq$ from $H$, giving that $M_\preceq \subseteq E_H^-$.

It remains to show that $E_H^- \subseteq M_\preceq$. Suppose otherwise. Then there exists a negative edge $(a{\to}b) \in E_H^- \setminus M_\preceq$. Then by definition of $M_\preceq$, there must be a negative edge $(x{\to}y)$ such that (i) $(a{\to}b)$ and $(x{\to}y)$ make up a primitive pair, and (ii) $(x{\to}y) \preceq (a{\to}b)$. Then $(a \to b)$ will be removed from $H$ when the algorithm considers $(x{\to}y)$ in line 4, contradicting that $(a{\to}b) \in E_H^-$. $\qquad\square$

## C    Proofs for Transitivity Bias

**Proposition 4.** *If $E_\theta$ is an energy function with transitivity bias, then for any $\theta$ there exists a $T \geq 0$ such that the digraph with edges $\{(u,v) \mid E_\theta(u,v) \leq T\}$ is transitively closed.*

*Proof.* Let $(u,v) \in E^{\mathrm{tc}}$. Then there exists a directed path $u := w_1 \to \cdots \to w_N =: v$ in $E$, hence $E_\theta(w_n, w_{n+1}) \leq T$ for all $n \in [\![N-1]\!]$. Let $k \in [\![N-2]\!]$, and assume we have $E_\theta(w_1, w_{k+1}) \leq T$. Since $E_\theta(w_{k+1}, w_{k+2}) \leq T$ as well, we apply the definition of transitivity bias to find $E_\theta(w_1, w_{k+2}) \leq T$. Thus, by induction, we have $E_\theta(w_1, w_N) = E_\theta(u,v) \leq T$. $\qquad\square$

**Proposition 5.** *Let $G$ be a transitively-closed digraph, $E_\theta$ an energy function with transitivity bias, and $H = (V, E_H^+, E_H^-)$ a sidigraph which distinguishes $G$ among transitively closed digraphs. If $T$ is the threshold associated with the transitivity bias for $\theta$, $E_\theta(u,v) \leq T$ for all $(u,v) \in E_H^+$, and $E_\theta(u,v) > T$ for all $(u,v) \in E_H^-$, then $\theta$ represents the digraph $G$.*

*Proof.* Let $E' = \{(u,v) \mid E_\theta(u,v) \leq T\}$, then Proposition 4 implies that the digraph $G' = (V, E')$ is transitively closed. Due to the assumptions on the values of the energy we have $E_H^+ \subseteq E'$ and $E_H^- \subseteq \overline{E'}$. Since $H$ is sufficient to distinguish transitively-closed digraphs, we have $G' = G$, as desired. $\qquad\square$

# D  Results with Error Bars

| Model | Training Edges | | Graph Type | | |
|---|---|---|---|---|---|
| | Pos. | Neg. | BT | nCRP | Price |
| GT-Box | ◐ | ◐ | 0.568 ± 0.187 | 0.796 ± 0.250 | 0.582 ± 0.240 |
| | ◐ | ● | 0.658± 0.103 | 0.775±0.131 | 0.602±0.147 |
| | ● | ◐ | **0.842**±0.154 | **0.904**± 0.058 | 0.849±0.112 |
| | ● | ● | 0.744± 0.159 | 0.877±0.101 | **0.909**±0.042 |
| SIM-VEC | ◐ | ◐ | 0.072±0.018 | 0.084±0.071 | 0.158±0.172 |
| | ◐ | ● | 0.180±0.065 | 0.285±0.112 | 0.336±0.126 |
| | ● | ◐ | 0.098±0.034 | 0.158±0.172 | 0.300±0.218 |
| | ● | ● | 0.508±0.039 | 0.409±0.023 | 0.494±0.151 |

# E  Hyperparameter Tuning

The experiments were run in a single-node, single-GPU setup on a cluster with the following GPU architectures:

- NVIDIA Tesla M40 (24GB VRAM)
- NVIDIA GeForce GTX TITAN X (12GB VRAM)
- NVIDIA GeForce GTX 1080 Ti (11GB VRAM)
- NVIDIA RTX 2080ti (11GB VRAM)
- NVIDIA Quadro RTX 8000 (48GB VRAM)

## E.1  Synthetic Graphs

To ensure the competitiveness of all experimental configurations for the synthetic graphs, we tune the learning rate and negative weight $\lambda_{\mathrm{neg}}$ for each graph independently by running Bayesian hyperparameter optimization with W&B Biewald et al. [2020] and for $E^{\mathrm{tc}}$ taking the values that yielded the best F1 at the end of 12 epochs, which we empirically observed corresponds to the "elbow" of many per-epoch F1 plots and thus gives a good indication of which settings are resulting in fast convergence. For $E^{\mathrm{tr}}$, we scaled the number of epochs to 60, since the number of training examples is substantially smaller for this setting than for transitive closure.

For each of the final experiment runs, we fetch the best learning rate and $\lambda_{\mathrm{neg}}$ and run for a full 40 epochs for $E^{\mathrm{tc}}$ or 200 for $E^{\mathrm{tr}}$, recording F1 at every epoch to produce a convergence plot.

## E.2  MeSH 2020

Since MeSH has at least thrice the number of nodes as any of the synthetic graphs in our experiments, it was infeasible to tune hyperparameters over all the same settings as for the smaller synthetic graphs. We therefore fix positive edge set $E^{\mathrm{tc}}$ (which empirically performs better than $E^{\mathrm{tr}}$), as well as the resource-efficient negative ratio $k = 4$, and do a Bayesian hyperparameter search for learning rate and negative rate $\lambda_{\mathrm{neg}}$ using W&B over the cross-product of model type (SIM-VEC vs GT-Box) and negative edge set ($\overline{E}$ vs $E_{H^*}$).

# F  Graph Family Hyperparameters and Edge Statistics

| Graph Type | $\Theta$ | $|V|$ | $|E|$ | $|E^{\text{tc}}|$ | $|E^{\text{tr}}|$ | $|\overline{E}|$ | $|E_{H^*}|$ | $|E|/|\overline{E}|$ | $1 - |E^{\text{tr}}|/|E^{\text{tc}}|$ | $1 - |E_{H^*}|/|\overline{E}|$ |
|---|---|---|---|---|---|---|---|---|---|---|
| Balanced Tree | $b = 2$ | 8192 | 90127 | 90127 | 8191 | 67010545 | 49153 | 0.1345% | 90.91% | 99.93% |
| | $b = 3$ | 8192 | 60620 | 60620 | 8191 | 67040052 | 86277 | 0.0904% | 86.49% | 99.87% |
| | $b = 5$ | 8192 | 44271 | 44271 | 8191 | 67056401 | 148201 | 0.066% | 81.5% | 99.78% |
| | $b = 10$ | 8192 | 31534 | 31534 | 8191 | 67069138 | 262765 | 0.047% | 74.02% | 99.61% |
| nCRP | $\alpha = 10$ | 10997.4 | 40373.6 | 40373.6 | 10996.4 | 120893115.0 | 1019607.5 | 0.0334% | 72.75% | 99.16% |
| | $\alpha = 100$ | 9713.8 | 22090.4 | 22090.4 | 9712.8 | 94326745.4 | 4035563.3 | 0.0234% | 56.03% | 95.72% |
| | $\alpha = 500$ | 9277.4 | 17389.5 | 17389.5 | 9276.4 | 86043713.3 | 11805777.4 | 0.0202% | 46.66% | 86.28% |
| Price | $c = 0.01, \gamma = 1.0, m = 1$ | 8192 | 8669.6 | 8669.6 | 8191.0 | 67092002.4 | 62675524.2 | 0.0129% | 5.43% | 6.58% |
| | $c = 0.01, \gamma = 1.0, m = 5$ | 8192 | 50281.7 | 50281.7 | 10294.3 | 67050390.3 | 27675479.7 | 0.075% | 79.52% | 58.72% |
| | $c = 0.01, \gamma = 1.0, m = 10$ | 8192 | 99878.0 | 99878.0 | 13940.9 | 67000794.0 | 22324190.0 | 0.1491% | 86.08% | 66.68% |
| | $c = 0.1, \gamma = 1.0, m = 1$ | 8192 | 13834.2 | 13834.2 | 8191.0 | 67086837.8 | 31970202.7 | 0.0206% | 40.3% | 52.35% |
| | $c = 0.1, \gamma = 1.0, m = 5$ | 8192 | 75013.5 | 75013.5 | 16418.3 | 67025658.5 | 6960419.4 | 0.1119% | 78.05% | 89.62% |
| | $c = 0.1, \gamma = 1.0, m = 10$ | 8192 | 138711.9 | 138711.9 | 23868.4 | 66961960.1 | 4171970.3 | 0.2072% | 82.79% | 93.77% |
| MeSH 2020 | N/A | 29934 | 40793 | 274437 | 40156 | 895739985 | 1953539 | 0.0046% | 85.37% | 99.78% |

Table 1: Statistics for the synthetic transitively-closed DAGs used in our experiments (Section 6), counts for the number of positive/negative edges in the equivalent and optimal distinguishing sidigraphs, and the ratios for the "full" and "reduced" experimental settings. While graphs under Balanced Tree are deterministic, the values of each entry for nCRP and Price are averaged over 10 random seeds.

# G Plots for $k = 128$

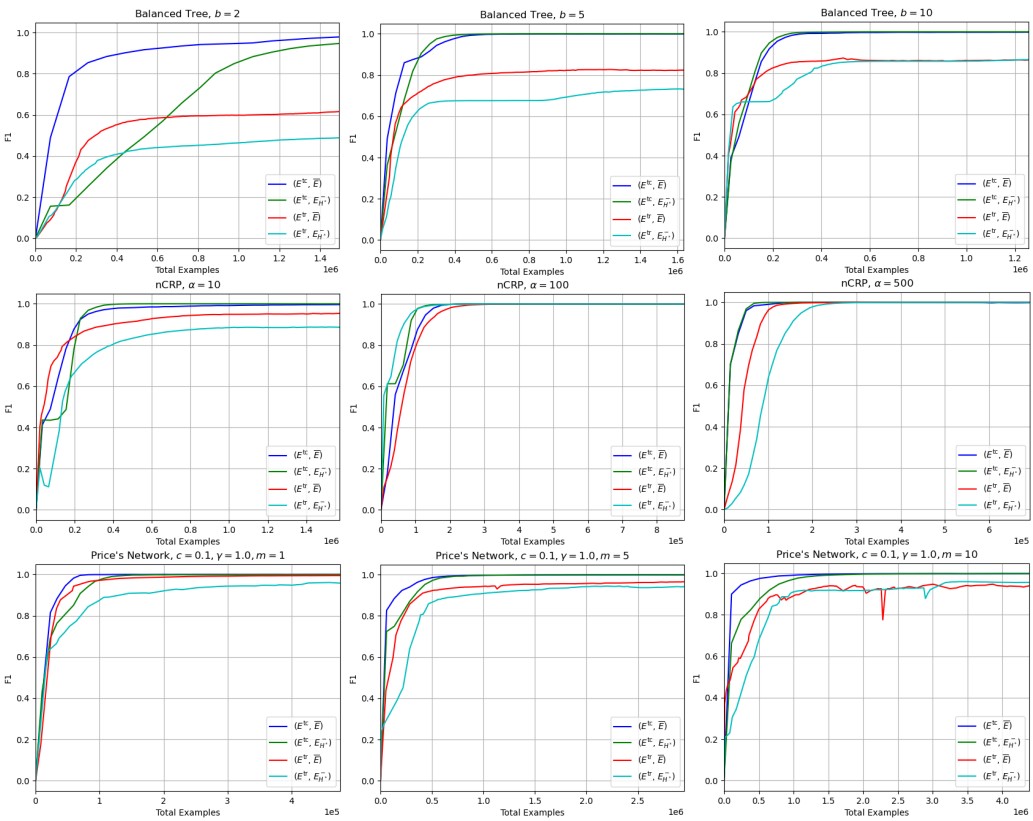

Figure 11: The plots show convergence of GT-Box for negative ratio $k = 128$. The top row shows the plots for balanced trees with branching factors $b = 2$, 5 and 10. The middle row for nCRP graphs with $\alpha = 10$, 100, and 500, respectively going left to right. The bottom row shows the plots for Price's graph with $c = 0.1$, $\gamma = 1.0$ and three values of $m = 1$, 5, and 10. The number of vertices in each graph is $\approx 2^{13}$.

