# OpenReview forum: "Learning Representations for Hierarchies with Minimal Support"
_NeurIPS.cc/2024/Conference — NeurIPS 2024 poster_

### Official Review · Reviewer_TmQr · 2024-07-11

**Soundness:** 3
**Presentation:** 2
**Contribution:** 2
**Rating:** 5
**Confidence:** 2

**Summary:**

This paper develops a framework to identify a subset of entries required to uniquely distinguish a graph among all transitively-closed DAGs. It achieves robust performance on synthetic hierarchies and a larger real-world taxonomy.

**Strengths:**

S1: A framework is proposed for detecting a sufficient smallest subset of entities in a graph adjacency matrix to specify a digraph.

S2: Details on formulating the problem are provided to help characterize the proposed method.

S3: A series of experimental studies were conducted to show the effectiveness of the proposed method.

**Weaknesses:**

W1:  The end-to-end efficiency is not clearly evaluated in this work, making it a bit unclear to assess the significance of the work in practice. For example, the proposed technique helps learn representations with minimal entries supported during training, where the minimal entries lead to reduced complexity in terms of computation; however, the hierarchy-aware sampling process itself adds additional operations, which may or may not result in increased overall complexity.

W2: The hierarchy aware sampling is a key technique in the proposed solution. The authors should elaborate the sampling process. The loss function provided in Section 5 seems relying on several not-well-defined terms, making it a bit concerned about the repeatability of the work.

W3: The presentation should be improved. For example, in Section 5,  the verb is missing in the first sentence. I assume the authors mean “formally —> formulate”.

**Questions:**

Refer to the weakness section above.

**Limitations:**

The authors called out the limitations in term of extending the proposed method to new combinations of graph properties and inductive biases, as well as that  the efficacy can vary when applying the method to graphs not transitively closed.

---

> ### Author Rebuttal · Authors · 2024-08-06
>
> Thank you for your review and questions.
>
> > W1: The end-to-end efficiency is not clearly evaluated in this work, making it a bit unclear to assess the significance of the work in practice. For example, the proposed technique helps learn representations with minimal entries supported during training, where the minimal entries lead to reduced complexity in terms of computation; however, the hierarchy-aware sampling process itself adds additional operations, which may or may not result in increased overall complexity.
> >
>
> All of the additional steps introduced by the hierarchy-aware sampling can be lumped under a one-time preprocessing procedure, as described in Algorithm 1.
>
> Since the total number of edges — both positive and negative — in the sidigraph is $O(|V|^2)$, each set of nested for-loops in Algorithm 1 (lines 2-5 as well as lines 6-9) takes $O(|V|^2)$ time, since the cost of removing a redundant edge is $O(1)$.
>
> In practice, the preprocessing step of FINDMINDISTINGUISHER can be run on CPU and can be parallelized w.r.t. the negative edges. Moreover, as evidenced in Table 1 Appendix H, the reduction in negative edges is often over 99%. Meanwhile, Figure 5 (GT-Box, rows 3-4) show that the convergence rates of Balanced Tree and nCRP (the graphs with the greatest negative reductions contributed by the reduced negative edges $E_{H}^{-}$) are higher using $E_{H}^{-}$ than using the full negative edge set $\overline{E}$. We therefore believe that our setup is useful in practice.
>
> > W2: The hierarchy aware sampling is a key technique in the proposed solution. The authors should elaborate the sampling process. The loss function provided in Section 5 seems relying on several not-well-defined terms, making it a bit concerned about the repeatability of the work.
> >
>
> To elaborate, the terms $\ell^{+}(x)=\lambda_{pos} x$ and $\ell^{-}(x) = -\lambda_{neg}~x$ in the loss equation of Section 5 are the same as the terms in Section 2.2 — positive and negative scalars which we call $\lambda_{pos}$ and $\lambda_{neg}$, respectively, to weight the energy contribution. We always set $\lambda_{pos}=1$ and we optimize for the best $\lambda_{neg}$ (please refer to Appendix G.1 and G.2). Therefore, for SimVec, the expanded loss function is:
>
> $\sum_{(u,v) \in E_{H}^{+}}\ell^{+}(E_{\theta}(u,v)) + \sum_{(u,v) \in E_{H}^{-}}\ell^{-}(E_{\theta}(u,v))$
>
> $=\sum_{(u,v) \in E_{H}^{+}}\ell^{+}(-\log\sigma(\theta_u \cdot \theta_v)) + \sum_{(u,v) \in E_{H}^{-}}\ell^{-}(-\log\sigma(\theta_u \cdot \theta_v))$
>
> $=\sum_{(u,v) \in E_{H}^{+}}(-\log\sigma(\theta_u \cdot \theta_v)) -\lambda_{neg}\sum_{(u,v) \in E_{H}^{-}}(-\log\sigma(\theta_u \cdot \theta_v))$
>
> (The energy function for SimVec is mentioned in line 236 of the paper.) For GT-Box, the expanded loss function is the same, except that the $-\log\sigma(\theta_u \cdot \theta_v)$ is replaced by the GT-Box energy function described in lines 181-185.  We hope that this alleviates your concerns regarding reproducibility! We will also provide a link to our implementation upon acceptance.
>
> > W3: The presentation should be improved. For example, in Section 5, the verb is missing in the first sentence. I assume the authors mean “formally —> formulate”.
> >
>
> Thank you for these observations — we will take care to improve and proof check our writing!

---

> > ### Comment · Reviewer_TmQr · 2024-08-09
> >
> > Thank you to the authors for addressing the review comments and providing additional details, including the loss formulation. I will review the rebuttal carefully and consider these details for further evaluation to determine if any changes to the score are warranted.

---

> > > ### Comment · Reviewer_TmQr · 2024-08-13
> > >
> > > Thanks again for the authors' efforts on addressing my comments. According to the complexity analysis in the rebuttal, the concern remains as in reality many graphs have very skewed degree distributions, which imply possible efficiency challenges around super nodes. The additional details for the hierarchy aware sampling improves the clarity a bit, but the novelty is not very clear given the key role of hierarchy aware sampling in the proposed solution. Overall, I would like to maintain my opinion on this paper.

---

> > > > ### Author Response · Authors · 2024-08-13
> > > > **Thank you for reviewing our rebuttal!**
> > > >
> > > > Thank you for reviewing our rebuttal!
> > > >
> > > > > According to the complexity analysis in the rebuttal, the concern remains as in reality many graphs have very skewed degree distributions, which imply possible efficiency challenges around super nodes.
> > > > >
> > > >
> > > > Indeed, some graphs are not as benefitted by the reduction in FINDMINDISTINGUISHER as other (compare the moderate reductions in Price graphs, Table 1, Appendix H, vs the drastic reduction in Balanced Trees). However, **on a large real-world hierarchy such as MeSH, the provably minimal support set does indeed reduce the negative support set by over 99% (cf. Table 1), while converging faster than the full negative support set (cf. Figure 8)**. Thus, our graph-theoretically optimal reduction is impactful in a real-world setting. We have also experimentally validated our theoretical findings on 3 distinct families of transitively-closed DAGs to yield a diverse suite of synthetic graphs. Could you please let us know if you have a particular graph or graph family in mind?
> > > >
> > > > > The additional details for the hierarchy aware sampling improves the clarity a bit, but the novelty is not very clear given the key role of hierarchy aware sampling in the proposed solution.
> > > > >
> > > >
> > > > We stress that **the key novelty of this paper is the minimal size of the support set rather than the sampling procedure**. Indeed, we use the same sampling procedure as in Boratko et al. 2021, and demonstrate that it yields comparable or better convergence even if sampling out of a support set of negative edges that is reduced by as much as 99%. Therefore, **we intentionally kept the sampling procedure the same between lines 74-75 and 211-212, in order to compare the impact of the respective support sets on convergence**.

---

> > > > > ### Comment · Reviewer_TmQr · 2024-08-14
> > > > >
> > > > > Hi authors, thanks for the response and sharing your thoughts. I believe I was asking about the impact of the super nodes of the original input graph. If I am correct, the high reduction rate of the negative support set is related to the equivalent sidigraph, conceptually introduced by the proposed method, rather than the original structural density or node degrees of the input graph. Since the equivalent sidigraph can be significantly denser (orders of the magnitude) than the input graph structure, the high reduction rate of negative support set per se would not convincingly address the concern regarding the super nodes in large scale graphs. Hence, I would like to maintain my opinion.

---

### Official Review · Reviewer_VFTZ · 2024-07-11

**Soundness:** 2
**Presentation:** 1
**Contribution:** 1
**Rating:** 3
**Confidence:** 2

**Summary:**

This paper proposes to distinguish a directed graph (digraph) among all transitivity-closed DAGs by finding minimal signed directed graph (sidigraph).
This paper exploits this idea to propose a more efficient algorithm for node embedding models.

**Strengths:**

- Theoretical supports in Sec. 3 for the sidigraph obtained by algorithm 1.

**Weaknesses:**

- It is not clear regarding prop 2 if the distinguishing serigraph H' is "lighter" than the transitive reducted G'.

This seems to "shift" the problem from reducing G to G' to reducing G to H'.
I do not see the benefit of obtaining H' over directly obtaining G'.

- The scope of this paper is unclear. In the introduction, the key question is bolded and may be answered Prop. 2. However, what is the rest of the paper?

- It is difficult to understand the aim of the experiments. In L253, the authors wrote as

> We investigate the impact of the available positive and negative support sets on model training; since the representational capacities of vector and box models are well-studied, we are not interested in the best F1 score attainable by these models, but the effect of respective support sets on convergence

However, the very first sentence of the result in L260 is

> First note that GT-BOX universally outperforms SIM-VEC on the graph modeling experiments

If the authors want to show that the proposed algorithm outperforms the existing one, then probably the authors want to compare with many more. If not, what is this sentence? I think that this experiment aims to show the superiority of the proposed algorithm by comparing it with SIM-VEC from the viewpoint of the effect of respective support sets on convergence. Thus, I think the authors want to compare with more.

This may be the presentation issue or content-wise issue. Anyhow, I hope the authors clarify these points.

**Questions:**

I hope the authors clarify the points raised in the weaknesses section.

---

> ### Author Rebuttal · Authors · 2024-08-06
>
> Thank you very much for your questions and comments.
>
> > It is not clear regarding prop 2 if the distinguishing serigraph H' is "lighter" than the transitive reducted G'.
> >
>
> Please note that while $G^{\prime}$ is a digraph with only positive edges specified, $H^{\prime}$ is a sidigraph with positive edges *as well as explicit negative edges* specified. Thus, if the positive edges of$H^{\prime}$ are already transitively-reduced, then indeed $H^{\prime}$ is not lighter than $G^{\prime}$ *in terms of positive edges.* However, this is not the case *in terms of the negative edges* of the equivalent sidigraph to $G^{\prime}$.
>
> Note that for $H=(V, E_{H}^{+}, E_{H}^{-})$ at the start of Proposition 2, since we are given that $H$ is a distinguisher of $G^{\prime}$, we are guaranteed to have $E_{H}^{+} \subseteq E$ and $E_{H}^{-} \subseteq \overline{E}$ by Definition 2. Therefore, removing any negative edge from $H$ to produce $H^{\prime}$ will necessarily make $H^{\prime}$ lighter than $H$, i.e., lighter than the equivalent sidigraph to the transitively-reduced $G^{\prime}$.
>
> For empirical evidence that $H^{\prime}$ is lighter than the equivalent sidigraph $H$ in terms of negative edges, please refer to Table 1 of Appendix H, where the rightmost column gives the reduction in negative edges contributed by Algorithm 1 to be as high as 99% for many graph families.
>
> > This seems to "shift" the problem from reducing G to G' to reducing G to H'. I do not see the benefit of obtaining H' over directly obtaining G'.
> >
>
> As noted above, $G^{\prime}$ can only inform us of the reduction in *positive edges*, and is ambiguous w.r.t. the reduction in negative edges. For the machine learning setting that is explored in the second part of this paper, we need to sample both positive and negative examples (edges) for training — therefore it is not enough to specify $(V, E^{tr})$ as it can imply $(V, E^{tr}, \overline{E})$, which offers no reduction in negative edges for the sampling pool. Reducing $G$ to $H^\prime$ implies explicitly reducing it to $(V, E^{tr}, E_{H}^{-})$, which may have orders of magnitude fewer negative edges from which to sample.
>
> > The scope of this paper is unclear. In the introduction, the key question is bolded and may be answered Prop. 2. However, what is the rest of the paper?
> >
>
> Indeed, the graph-theoretic question posed by our paper is answered by Proposition 2 and Algorithm 1 which is based on it. However, we would like to understand whether the provable graph-theoretic sufficiency of $H^{*}$ can be exploited by energy-based node embedding models.
>
>
> Proposition 4 shows that if the energy-based node embedding model has transitivity bias, then there exists a parameter configuration that can unambiguously represent $G$  if it can represent a distinguisher of $G$ (such as $H^*$). However, the question of whether this configuration is achievable in practice is an empirical one, which our experiments section explores or real data. The experiment section shows that training the transitivity-biased GT-Box with positive and negative edges from $H^*$ achieves the configuration, whereas training SimVec (which does not have transitivity bias) while sampling from the same support set results in very poor performance (see Figure 6, $(E^{tr}, E_{H}^{-})$).
>
> > It is difficult to understand the aim of the experiments. In L253, the authors wrote as
> >
> >
> > > We investigate the impact of the available positive and negative support sets on model training; since the representational capacities of vector and box models are well-studied, we are not interested in the best F1 score attainable by these models, but the effect of respective support sets on convergence
> > >
>
> > However, the very first sentence of the result in L260 is
> >
> >
> > > First note that GT-BOX universally outperforms SIM-VEC on the graph modeling experiments
> > >
>
> > If the authors want to show that the proposed algorithm outperforms the existing one, then probably the authors want to compare with many more. If not, what is this sentence?
> >
>
> Thank you for pointing this out, and we see how this sentence can be misleading as to the aim of our experiments. Indeed, **the main aim is to understand the impact of {$E^{tc}$,  $E^{tr}$} and of {$\overline{E}, E_{H}^{-}$} on convergence.** Thus, we were trying to emphasize that it is not the best achievable F1 that is of interest to us, but how the convergence is affected by the support sets. **The fact that SimVec underperformed GT-Box w.r.t. the ultimate F1 (as written in L260) does not obscure the pervasive trend that GT-Box is consistently able to benefit from the hierarchy-aware negative edge set $E_{H}^{-}$, while the performance of SimVec plummets with $E_{H}^{-}$ to the point where SimVec is not able to learn anything useful.** We will certainly revise the first sentence of 6.1 for the camera-ready version, and we appreciate your suggestion!
>
> > I think that this experiment aims to show the superiority of the proposed algorithm by comparing it with SIM-VEC from the viewpoint of the effect of respective support sets on convergence. Thus, I think the authors want to compare with more.
> >
>
> We note that GTBox is a generalization of a number of models that have transitivity bias. Meanwhile, SimVec reflects a simple but widespread vector-based node embedding model that does not have transitivity bias. Based on Proposition 4, we have strong evidence to believe that analogous models without transitivity bias would exhibit similar poor trends to SimVec when coupled with $E_{H}^{-}$.

---

> > ### Comment · Reviewer_VFTZ · 2024-08-11
> > **Thank you very much for the rebuttal**
> >
> > Thank you for your effort on the rebuttal.
> >
> > While I think that motivation for the first part is well presented and the results are very sound, the second part is largely improved.
> >
> > The authors may want to more explain on the motivation behind the second part in introduction, as in the similar manner as the first part. Also, as far as I understand, comparing two on transitivity bias is not enough. Is the GT-Box is the ONLY generalized model that has the transitivity bias? Is there any model that has a transitivity bias other than the family of GT-Box? Is there any other possible baseline other than sim-vec? Why you can say that the transition bias has effects on convergence only comparing two, where other factors than transitivity bias may involve?
> >
> > Due to this concern, I keep my score.

---

> > > ### Author Response · Authors · 2024-08-12
> > > **Thank you for reviewing our rebuttal!**
> > >
> > > Thank you for reviewing our rebuttal!
> > >
> > > > The authors may want to more explain on the motivation behind the second part in introduction, as in the similar manner as the first part.
> > > >
> > >
> > > The motivation for the second part is to demonstrate that the graph-theoretic results can be applied in practice, by leveraging a machine learning model with transitivity bias. (See also lines 41-45 in the introduction).
> > >
> > > > Also, as far as I understand, comparing two on transitivity bias is not enough. […]
> > > >
> > >
> > > We selected models for empirical evaluation with two goals in mind:
> > >
> > > ------------------
> > >
> > > 1. Validate that our theoretical claims can be obtained in the practical setting, in the context of things such as hyperparameter selection and stochasticity in the training loop.
> > >
> > > To do so, it was necessary to demonstrate that a model with transitivity bias could achieve similar performance when training on our proposed reduced edge set (up to 99.9% reduction) as when trained on the full edge set. As such, we needed to select a model which (a) had strong modeling capacity on transitively closed DAGs, and (b) satisfied the formal definition of transitivity bias, as in Definition 4. Box embeddings were the obvious choice to satisfy (a), having the best performance of any model on transitively closed DAGs as reported in Boratko et al. 2021. We then proved that box embeddings satisfy (b) in Proposition 5. We then tested the model, and observed that it was, indeed, able to obtain similar or better results when using our reduced edge set.
> > >
> > > This, on its own, is enough to validate that the theoretical claims at least can be obtained in practice.
> > >
> > > ------------------
> > >
> > > 2. Empirically demonstrate that the facet of transitivity bias is important.
> > >
> > > For this purpose, we needed to choose a model which (a) had strong modeling capacity on transitively closed DAGs, but (b) did not satisfy the formal definition of transitivity bias from Definition 4. Again, SimVec is the logical choice here based on the results from Boratko et al. 2021 - it obtains perfect performance on modeling transitively closed DAGs (once given sufficient dimensionality), however can easily be shown to not satisfy Definition 4. The substantial performance gap on SimVec when using the reduced set of edges supports our conclusion.
> > >
> > > ------------------
> > > In particular, we do not intend these empirical results to justify a universal claim regarding all models which have or do not have transitivity bias - such a claim is already supported by the theoretical results. Rather, our intention is for these empirical results to demonstrate an existential claim that the theoretical results can be leveraged in practice.
> > >
> > > A complete analysis of existing embedding models and the extent to which they have or do not have transitivity bias is actually rather subtle and outside the scope of this work. First, while geometric intuition suggests that it is likely that any region-based model (such as Order Embeddings or disks) would have transitivity bias in the sense of Definition 4, it ultimately depends on technical details related to the energy function. In particular, the proof for box embeddings does not trivially translate to the case of Hyperbolic Entailment Cones. Second, there are more subtle distinctions (e.g. a notion of “weak transitivity bias”, wherein the set of parameters for which the energy function can be thresholded to yield a transitively-closed DAG is sufficiently large) which would be relevant for such a detailed analysis.

---

### Official Review · Reviewer_gCXR · 2024-07-13

**Soundness:** 3
**Presentation:** 3
**Contribution:** 3
**Rating:** 7
**Confidence:** 4

**Summary:**

This paper proposes a novel framework for identifying a minimal subset of entries in the adjacency matrix that uniquely distinguishes a directed acyclic graph (DAG) among all transitively-closed DAGs. The authors provide a provably optimal algorithm for computing this minimal set. They then leverage these insights to develop hierarchy-aware sampling, which allows training node embedding models more efficiently by focusing only on the essential graph information. Specifically, they prove that models with an inductive bias of transitivity, such as box embeddings, can learn faithful representations of hierarchies using substantially fewer training examples selected by their sampling approach. Experiments on synthetic and real-world graphs demonstrate that their method significantly reduces training data by up to 99% while maintaining or improving model performance and convergence rates compared to uniform negative sampling.

**Strengths:**

- The theoretical grounding of the paper looks good to me, with key properties of the proposed framework and algorithms formally stated and proven. The experiments systematically evaluated the impact of algorithm design choices (e.g. positive/negative edge sets) and data characteristics (e.g. graph structures) on model performance.
- Learning faithful graph representations with minimal data has important computational benefits and is conceptually valuable for understanding the essential structural information needed to distinguish different graphs. The reduction in training data achieved by the proposed sampling method is substantial. The theoretical framework and practical techniques in this paper could be built upon to develop further improved graph representation learning approaches.

**Weaknesses:**

- While the paper demonstrates the effectiveness of transitivity bias and hierarchy-aware negative sampling for DAGs, the authors could discuss whether these ideas extend to learning representations of other graph families characterized by different structural properties. Are there other useful inductive biases worth incorporating into node embeddings and corresponding "structure-aware" sampling strategies?
- The experiments focus on evaluating hierarchy-aware sampling, but there is less empirical analysis of the FINDMINDISTINGUISHER algorithm itself, e.g. runtime complexity, actual sparsity of computed sidigraphs, etc. Knowing when the sidigraph is substantially smaller than the original graph is important for determining when hierarchy-aware sampling is computationally beneficial.

**Questions:**

- The framework of sidigraphs for distinguishing graphs with a given property seems quite general. Have the authors considered instantiating it for properties other than transitivity?
- Do the authors have intuition or theory on why hierarchy-aware sampling provides a greater speed-up on some graphs than others (e.g. balanced trees vs Price graphs)?
- Could hierarchy-aware sampling also improve the computational efficiency of non-embedding GNN approaches for learning hierarchical structures, since they also implicitly perform a form of negative sampling via contrastive estimation?

**Limitations:**

Yes, there is a section about the limitations of the work.

---

> ### Author Rebuttal · Authors · 2024-08-06
>
> Thank you for your encouraging review and questions!
>
> > While the paper demonstrates the effectiveness of transitivity bias and hierarchy-aware negative sampling for DAGs, the authors could discuss whether these ideas extend to learning representations of other graph families characterized by different structural properties. Are there other useful inductive biases worth incorporating into node embeddings and corresponding "structure-aware" sampling strategies?
> >
>
> > …The framework of sidigraphs for distinguishing graphs with a given property seems quite general. Have the authors considered instantiating it for properties other than transitivity?
> >
>
> In this work we have focused on transitivity, but we are certainly on the lookout for other interesting structural properties for which this framework would be useful. One extension we think it would be interesting to explore is relation composition on a multi-relational graph - however, such an exploration would fall outside of the scope of this paper.
>
> > The experiments focus on evaluating hierarchy-aware sampling, but there is less empirical analysis of the FINDMINDISTINGUISHER algorithm itself, e.g. runtime complexity, actual sparsity of computed sidigraphs, etc. Knowing when the sidigraph is substantially smaller than the original graph is important for determining when hierarchy-aware sampling is computationally beneficial.
> >
>
> Thank you for your suggestion. Since the total number of edges — both positive and negative — in the sidigraph is $O(|V|^2)$, each set of nested for-loops in Algorithm 1 (lines 2-5 as well as lines 6-9) takes $O(|V|^2)$ time, since the cost of removing a redundant edge is $O(1)$. Moreover, in practice, removing a negative edge in the inner loop will often reduce the number of edges left to iterate over in the outer loop. We will make sure to include this runtime analysis in the camera-ready version.
>
> For detailed statistics about sidigraph sparsity please refer to Table 1 in Appendix H, where we provide the unpruned number of positive and negative edges of each graph we consider, as well as the ratios of the “reduced” to the “full” sets of positive/negative/total edges. The rightmost column in Table 1 demonstrates that for Balanced Tree, nCRP and MeSH, the reduction in negative edges is drastic (at least 99% for one or more graph configurations of those families).
>
> > Do the authors have intuition or theory on why hierarchy-aware sampling provides a greater speed-up on some graphs than others (e.g. balanced trees vs Price graphs)?
> >
>
> While we do not have a formal proof concerning the influence of particular graph structures on convergence, we would like to refer the reviewer to Table 1 in Appendix H, which shows (in the rightmost column) that for Balanced Trees, the reduction in negative edges contributed by FINDMINDISTINGUISHER is always at least 99%, while it is much less for Price graphs. This suggests that balanced trees can benefit substantially more from the reduction than Price graphs.
>
> > Could hierarchy-aware sampling also improve the computational efficiency of non-embedding GNN approaches for learning hierarchical structures, since they also implicitly perform a form of negative sampling via contrastive estimation?
> >
>
> Yes, we agree that hierarchy-aware negative sampling could potentially be used to provide an adaptive noise distribution for training non-embedding GNN models which attempt to learn latent hierarchical structure. To expand on this slightly, while we apply our algorithm to a static DAG, this could also be applied to a dynamic graph, such as an “active learning” setting (eg. where the true graph is unknown and we query humans for the existence or non-existence of particular edges) or the setting you describe, where a model such as a GNN attempts to learn the latent hierarchical structure and the edges presented to the training algorithm update dynamically using our constructive algorithm based on the current model parameters. This is an interesting direction for future exploration!

---

> > ### Comment · Reviewer_gCXR · 2024-08-12
> > **Feedback on rebuttal**
> >
> > I thank the reviewers for their response to my review in their rebuttal. Several of their comments directly address the concerns I raised, while others acknowledge that certain points I highlighted represent potential avenues for future research. The paper presents interesting findings, and I believe the authors would benefit from further academic discussions on this research topic. In consideration of the aforementioned factors, I have determined to increase my assessment score.

---

### Official Review · Reviewer_eH8o · 2024-07-13

**Soundness:** 2
**Presentation:** 3
**Contribution:** 3
**Rating:** 6
**Confidence:** 4

**Summary:**

Authors propose an algorithm to identify a subset of entries required to uniquely distinguish a graph among all transitively-closed DAGs, based on the theoretical analysis of transitive reduction of the associated signed digraph. These newly identified subsets are leveraged to learn node embeddings via contrastive learning approaches based on negative sampling more efficiently. The relevance of this novel contrastive approach is empirically validated on several synthetic datasets of DAG and one real-world dataset, leading to comparable or better performances than vanilla negative sampling strategies.

**Strengths:**

- Overall the paper is well-written.
- Provide an algorithm to uniquely identify transitively-closed DAGs (Alg. 1) and prove its convergence.
- Theoretically analysis the relation between the transitive bias of an energy discriminating edges and their graph reduction technique.
- Study the transitive bias of box embeddings and their relations to bit vectors.
- Empirical study of various negative sampling strategy supporting the relevance of the methodology proposed by authors.

**Weaknesses:**

*edits after rebuttal in italic*

1. As expressed by authors the range of applications relating to their study remains significantly narrow even if interesting. It could be of interest to compare their approach on the studied datasets with methods enforcing transitivity while covering a broader score e.g Rot-Pro (Song & al, 2021 in the paper)

2. Some points remain unclear to me in the experiments:
   - a) *[partly adressed]* Could you further detail the architecture and optimization procedure followed in your experiments ? Results reported in Figure 1 (Boratko & al, 2021) with d=128 (as you set) seems to be significantly better than the ones you report for sim-vec, could you clarify this matter ? Their analysis also emphasize that it will be relevant to study the sensitivity w.r.t $d$ across various negative sampling strategy, i encourage you to do so. It would also be of interest to check if these rankings are consistent while using another transitive representation learning scheme like GT-box.
   - b) *[Done]* From L.245, I expected comparison between k=4 and 128 but only results for k=4 are reported in the paper. Could you explain why and potentially fix that ?
   - c) *[partly adressed]* Unclear analysis of results for synthetic dataset, potentially to be refined by studying correlations with some dataset statistics. For instance, why are there opposite dynamics w.r.t to the depth between 'balanced tree' and 'nCRP' ?
   - d) Experiments on real datasets stopped too early ? It could be of interest to pursue experiments on the real dataset reported in Figure 8 until specific models seem to have converged as dynamics with small number of samples seem more erratic.

small tipos:
- L.208, missing word ? we formally `explain`...
- L.37-38: misleading / big claim, as you study a very specific type of graph.

**Questions:**

I invite authors to discuss and address the weaknesses/questions I have mentioned above, so that I can consider improving my score.

**Limitations:**

The authors have made an effort to address the limitations of their work, I encourage them to answer the questions above to ensure that all potential limitations have been covered.

No potential negative societal impact.

---

> ### Author Rebuttal · Authors · 2024-08-06
>
> Thank you for your review and insightful questions!
>
> > 1. As expressed by authors the range of applications relating to their study remains significantly narrow even if interesting. It could be of interest to compare their approach on the studied datasets with methods enforcing transitivity while covering a broader score e.g Rot-Pro (Song & al, 2021 in the paper)
> >
>
> Thank you for sharing the Rot-Pro paper. It is an interesting work on KG embeddings, much like TransE, RotatE, etc. According to our understanding, Rot-Pro is designed for the task of KG completion, where a KG is a multi-relational graph, i.e. a graph with categorical edge properties. In Rot-Pro, the transitivity is encoded primarily in the relation representation. In contrast, our work focuses on the efficiency of learning representations for directed acyclic graphs — i.e. there is only one relation. Given the success of our approach, extending it to the case of multi-relational KG completion definitely seems interesting, but it is out of scope for the current paper.
>
> > 2a) Could you further detail the architecture and optimization procedure followed in your experiments ?
> >
>
> We describe the architecture of GT-Box in Section 4.2. For GT-Box, a node is parameterized by two $d$-dimensional corners. The GT-Box score function between two boxes gets computed as in the formula between lines 189-190, its energy function as in the formula between lines 182-183.
>
> Concerning optimization procedure, please refer to Appendix G for details on optimization procedure for Synthetic Graphs (G.1) and MeSH (G.2)
>
> > 2a) Results reported in Figure 1 (Boratko & al, 2021) with $d=128$ (as you set) seems to be significantly better than the ones you report for sim-vec, could you clarify this matter ?
> >
>
> Thank you for bringing up this point. We would like to note that we did not do hyperparameter tuning for learning rate and negative weight ($\lambda_{neg}$) with highest attainable F1 as the objective. Rather, we did hyperparameter tuning with fast convergence as the objective — we achieved this heuristically by cutting off training at the end of a fixed number of epochs which we observed corresponded to the “elbow” of the convergence curve, and taking the hyperparameters that yielded the highest F1 at that point. The learning rate and $\lambda_{neg}$ have therefore been optimized for the purposes of *fastest* convergence, and therefore may underperform on best F1, which was the metric optimized for in Boratko et al. 2021. We detail this hyperparameter optimization procedure for synthetic graphs in Appendix G.1 and for MeSH in Appendix G.2.
>
> > 2a) Their analysis also emphasize that it will be relevant to study the sensitivity w.r.t  across various negative sampling strategy, i encourage you to do so.
> >
>
> We would be open to experimenting with other negative sampling strategies if you can suggest some particular ones. We were unable to find other relevant negative sampling strategies in Boratko et al. 2021.
>
> > b) From L.245, I expected comparison between k=4 and 128 but only results for k=4 are reported in the paper. Could you explain why and potentially fix that ?
> >
>
> Apologies for the oversight. We are attaching a 1-page pdf of results for $k=128$ to the global response, and will include it in the appendix. Based on the plots, the advantages of hierarchy-aware negative sampling are more apparent in the low-resource $k=4$ than in the higher-resource $k=128$.
>
> > c) Unclear analysis of results for synthetic dataset, potentially to be refined by studying correlations with some dataset statistics. For instance, why are there opposite dynamics w.r.t to the depth between 'balanced tree' and 'nCRP' ?
> >
>
> Can you please clarify what you mean by opposite dynamics in those two graph families? Note that since all graphs are generated with (almost) the same number of nodes $|V|$, we always have a tradeoff between depth and breadth, which might confound some conclusions.
>
> > d) Experiments on real datasets stopped too early ? It could be of interest to pursue experiments on the real dataset reported in Figure 8 until specific models seem to have converged as dynamics with small number of samples seem more erratic.
> >
>
> We can certainly rerun this training for longer for the camera-ready version. However, we do not think that the trend will change after more epochs.
>
> > small tipos:
> >
>
> > L.208, missing word ? we formally `explain`...
> >
>
> > L.37-38: misleading / big claim, as you study a very specific type of graph.
> >
>
> Thank you — we will address these typos promptly!

---

> > ### Comment · Reviewer_eH8o · 2024-08-10
> > **Answer to author**
> >
> > Thank you for your rebuttal, some of my concerns have been correctly addressed. Could you please complete your rebuttal by answering my following questions in order to reach a final decision?
> >
> > > 2-a) Their analysis also emphasize that it will be relevant to study the sensitivity w.r.t $d$ across various negative sampling strategy.
> >
> > Sorry for the misunderstanding, I was essentially referring to studying the sensitivity w.r.t $d$ (d=64*2=128) in your experiments e.g $(d=64, 32..)$, with the different positive and negative sets as you did in the main paper.
> >
> > 2-b) Thank you for these additional experiments. Could you analyze these results e.g by comparing performances between $k=4$ and $k=128$? $E_{H^*}^-$ seems significantly less relevant in the second case than the first one.
> >
> > > 2-c) Unclear analysis of results for synthetic datasets ...
> >
> > From figure 7 and your descriptions, if I understand correctly, the smaller $b$ gets for balanced trees, the deeper they are and the less relevant the use of $E_{H^*}^-$ seems. Whereas for nCRP, the smaller $\alpha$, the deeper the hierarchies (in the same way as with b for trees), but the most relevant seems to be $E_{H^*}^-$. So could you explain these opposite dynamics ?

---

> > > ### Author Response · Authors · 2024-08-12
> > > **Thank you for your clarifications!**
> > >
> > > Thank you for your clarifications!
> > >
> > > > 2-a) sensitivity to $d$:
> > > >
> > >
> > > While we think the sensitivity to $d$ is indeed interesting to explore, we are not sure we understand the motivation for this study. We have set $d=64$ in our experiments by analogy to the best-performing setting of Boratko et al. 2021 (cf. Figure 1, row 3 in their paper) because we expected that setting to draw the contrast between regular and hierarchy-aware sampling most sharply, reducing confounding factors such as training instability, which could easily result from too small a $d$. Meanwhile, setting $d > 64$ seemed excessive, as $d=64$ already gives enough capacity for both the models to represent the graphs perfectly.
> > >
> > > Boratko et al. also show that setting $d=8$ (Figure 1, row 1) results in very poor performance for the SimVec baseline even with $\overline{E}$. For our experiments, we think this would obscure the salient trend of $E_{H^*}^{-}$ causing SimVec (a non-transitivity-biased baseline) to crash.
> > >
> > > > 2-b) k=4 vs k=128
> > > >
> > >
> > > We agree with your observation that $k=4$ appears to benefit more from $E_{H^*}^-$ than does $k=128$, for which the convergence curves for $\overline{E}$ and $E_{H^*}^-$ are closer together. We think this is to be expected, because the more negatives we sample from the unpruned pool $\overline{E}$, the more likely it is that we draw “high-signal” negative edges, i.e., the $E_{H^*}^-$ edges which we explicitly filter for using FINDMINDISTINGUISHER. Meanwhile, if we sample very few negative edges, drawing from an unpruned pool will likely result in fewer “high-signal” negative samples per positive example, and will take longer to converge. This trend actually makes our approach more attractive for larger graphs, where the distribution of $\overline{E}$ makes it unlikely to sample many high-signal edges using a small negative ratio $k$.
> > >
> > > > 2-c) Balanced tree vs nCRP
> > > >
> > >
> > > Thank you for clarifying this interesting observation. We would like to refer the reviewer to [Figure 4 in the appendix of Boratko et al. 2021](https://proceedings.neurips.cc/paper_files/paper/2021/file/88d25099b103efd638163ecb40a55589-Supplemental.pdf) for a visualization of a Balanced Tree vs an nCRP graph. We note that Balanced Tree graph is characteristically less random than the nCRP graph. Therefore we hypothesize that the reduced set of negatives $E_{H^*}^-$ is more uniformly informative for the Balanced Tree graphs than for the remaining nCRP graphs, and the change from $\alpha=100$ to $\alpha=500$ for the nCRP might not actually be a trend inverse to the observed Balanced Tree trend.

---

### Official Review · Reviewer_ShEU · 2024-07-15

**Soundness:** 3
**Presentation:** 2
**Contribution:** 3
**Rating:** 7
**Confidence:** 4

**Summary:**

This paper addresses the challenge of training node embedding models on large directed graphs (digraphs) where it is impractical to observe all entries of the adjacency matrix during training, necessitating sampling methods. Recognizing that many entries remain unobserved in very large digraphs, the authors develop a novel framework to identify a minimal subset of entries required to uniquely distinguish a graph among all transitively-closed directed acyclic graphs (DAGs). They provide an explicit algorithm to compute this minimal set and empirically demonstrate that training node embedding models with this subset improves efficiency and performance, assuming the energy function has an appropriate inductive bias. Experiments on synthetic hierarchies and a large real-world taxonomy show robust performance, with improved convergence rates and a reduction in training examples by up to 99%, making the method highly effective in resource-constrained settings.

**Strengths:**

1. This paper is well written. The notations are clear and the literature review is sufficient.
2. By formulating the disjoint subset of notes in a graph in the notion of signed digraph, the proposed FINDMINDISTINGUISHER method is able to optimally handle both acyclic graphs with minimal support.
3. Box Embeddings is used for node embedding, so that the embeded space has a differentiable structure.

**Weaknesses:**

1. The effciency and robustness of the proposed method is verified via experiments on various types of hierarchies, including Balanced tree, nCPR, Price, and MeSH. However, it could be of more intuitive if the generic motivation of pursuing hierarchy-awareness can be illustrated with real world applications.

**Questions:**

It would be appreciate if the author can illustrate the motivation of this work in the pursuit of effciency and robustness in learning representations for hierarchy with minimal support.

---

> ### Author Rebuttal · Authors · 2024-08-06
>
> Thank you for your encouraging review and questions!
>
> > 1. The effciency and robustness of the proposed method is verified via experiments on various types of hierarchies, including Balanced tree, nCPR, Price, and MeSH. However, it could be of more intuitive if the generic motivation of pursuing hierarchy-awareness can be illustrated with real world applications.
> >
>
> Thank you for this advice. One highly related application is multi-label classification where the label space forms a taxonomy, and where we model the labels by means of region-based embeddings (cf. e.g. Patel et al. 2022). If the embedding of an instance falls within a region corresponding to some fine-grained label (e.g. a *MacBook Pro*), we would like that label to be contained under all its parent labels in the taxonomy (e.g. *< Laptop < PC < electronics < object*). In large product hierarchies, jointly training the instance embedding model together with label embeddings can be a computationally intensive task, where the label-space training can be alleviated by hierarchy-aware sampling.
>
> A broader potentially fruitful application of hierarchy-awareness is active learning with a human-in-the-loop. As we mention in the Introduction lines 20-22, “*when obtaining annotations for edges of an unknown graph, the full adjacency matrix is unknown to us and we obtain the value of any particular entry by requesting an annotation*”. Building up a taxonomy from open-source human knowledge (and training corresponding embeddings for fast retrieval) can require many human annotations — and as we have observed in the case of MeSH, we can obtain the minimal distinguishing sidigraph with 99% fewer edges (annotations) than in its equivalent sidigraph (cf. rightmost column of Table 1, Appendix H).
>
> > It would be appreciate if the author can illustrate the motivation of this work in the pursuit of effciency and robustness in learning representations for hierarchy with minimal support.
> >
>
> Once again, we appreciate this advice and will certainly make the real-world motivations (such as the above-mentioned) more explicit in the introduction to make a stronger paper.

---

> > ### Comment · Reviewer_ShEU · 2024-08-12
> >
> > I have read the rebuttal and thank the authors for their candid responses.
> >
> > I maintain my positive opinion on this paper.

---

### Author Rebuttal · Authors · 2024-08-06

We are sharing a pdf with plots for GT-Box with negative ratio $k=128$ (analogous to Figure 7, which is for $k=4$).

---

### Author Response · Authors · 2024-08-07

We thank the reviewers for their comments, questions, and feedback. Multiple reviewers found the paper well-written (ShEU, eH80) and impactful (gCXR), with clear theoretical contributions (eH80, VFTZ, gCXR) and sufficient empirical support (eH80, gCXR, TmQr).
We hope to have fully addressed questions in the individual replies to each reviewer.

---

### Decision · Program_Chairs · 2024-09-25

**Decision:**

Accept (poster)

**Comment:**

This paper presents a novel procedure to identify a unique directed acyclic graph among all such possible graphs by querying the minimum possible potential edges (entries in the adjacency matrix). The authors utilize this framework to develop hierarchy-aware sampling for training node embeddings more efficiently, reducing the number of training samples required. They also report experiments on synthetic hierarchies and real-world taxonomies, demonstrating the efficiency of their method.

I think the authors propose both an interesting theoretical question, and evaluate practical impact from exploiting known structure of the underlying graph. I think this is an interesting avenue for research, and can motivate up several follow-up works. I recommend acceptance.